# Pro-ferroptotic signaling promotes arterial aging via vascular smooth muscle cell senescence

Di-Yang Sun [1,2,11], Wen-Bin Wu [1,11], Jian-Jin Wu [3,11], Yu Shi [4,11], Jia-Jun Xu [5,11], Shen-Xi Ouyang[4], Chen Chi [4,6], Yi Shi[7,8], Qing-Xin Ji [4], Jin-Hao Miao [9], Jiang-Tao Fu[1], Jie Tong[4], Ping-Ping Zhang [1], Jia-Bao Zhang [1,10], Zhi-Yong Li[1,10], Le-Feng Qu [3] ✉, Fu-Ming Shen [4] ✉, Dong-Jie Li [4] ✉ & Pei Wang [1,10] ✉

Senescence of vascular smooth muscle cells (VSMCs) contributes to aging-related cardiovascular diseases by promoting arterial remodelling and stiffness. Ferroptosis is a novel type of regulated cell death associated with lipid oxidation. Here, we show that pro-ferroptosis signaling drives VSMCs senescence to accelerate vascular $NAD^+$ loss, remodelling and aging. Pro-ferroptotic signaling is triggered in senescent VSMCs and arteries of aged mice. Furthermore, the activation of pro-ferroptotic signaling in VSMCs not only induces $NAD^+$ loss and senescence but also promotes the release of a pro-senescent secretome. Pharmacological or genetic inhibition of pro-ferroptosis signaling, ameliorates VSMCs senescence, reduces vascular stiffness and retards the progression of abdominal aortic aneurysm in mice. Mechanistically, we revealed that inhibition of pro-ferroptotic signaling facilitates the nuclear-cytoplasmic shuttling of proliferator-activated receptor-γ and, thereby impeding nuclear receptor coactivator 4-ferrtin complex-centric ferritinophagy. Finally, the activated pro-ferroptotic signaling correlates with arterial stiffness in a human proof-of-concept study. These findings have significant implications for future therapeutic strategies aiming to eliminate vascular ferroptosis in senescence- or aging-associated cardiovascular diseases.

Cellular senescence is an essential physiological process that is mainly designed to eliminate unwanted cells. In physiological conditions, senescent cells can be removed by the immune system, facilitating tumor suppression and wound healing. However, senescence is also activated upon pathological damage. Recently, senescence has emerged from humble beginnings as an in vitro phenomenon into recognition as an in vivo cellular hallmark of aging[1]. Vascular smooth muscle cells (VSMCs) undergo senescence upon various vascular stimuli such as DNA damage, genomic instability, telomere shortening, and oxidative stress. Failure of removal of senescent VSMCs promotes arterial stiffness, degradation, and remodeling, all of which are features of vascular aging and vascular aging-related cardiovascular disease (CVD)[2]. Moreover, the senescent VSMCs release a secretome, which is always called as senescence-associated secretory phenotype (SASP), that attracts immune cell infiltration into vasculature to induce pro-inflammatory microenvironment and calcification[2]. Thus, targeting senescent cells may reverse vascular senescence and extend lifespan in mice.

Recently, a growing body of literature has demonstrated that senescence, as well as aging, is controlled by intracellular metabolic states. Nicotinamide adenine dinucleotide (NAD⁺) is an important metabolic-active co-factor in cells by playing pivotal roles in intracellular metabolic state control[3]. NAD⁺ contributes to signaling transduction, post-translational modification, DNA repair, epigenetic modification, and cell fate via regulating NAD⁺-consuming enzymes, including sirtuins, poly-ADP-ribose polymerases (PARPs), and CD38/CD157 NADases[4]. Notably, intracellular NAD⁺ level declines with senescence and aging[5,6], while restoration of NAD⁺ homeostasis reverses vascular senescence[7] and aging[8–10].

Ferroptosis, a novel form of regulated cell death (RCD) characterized by overwhelming accumulation of ferrous iron (Fe²⁺)-driven lethal lipid reactive oxygen species (ROS), is morphologically, biochemically, and genetically distinct from other types of well-known cell death such as apoptosis and necrosis[11–13]. The main mechanism of biological toxicity of Fe²⁺ is mediated by the classical Fenton reaction, which produces highly active hydroxyl radicals to attack plasma membrane and promote cell death[11–13]. Although the precise signaling transduction has not been fully elucidated, it has been known that ferroptosis is regulated by multiple layers of signaling pathways and procedures, including anti-oxidant enzymes (especially glutathione peroxidase 4 [GPX4]), lipid peroxidation products, mitochondrial function, interferon-related genes, etc.[14]. Ferroptosis plays multiple roles in vital cellular events such as cell death[11–13] and tumor suppression[15]. Moreover, ferroptosis signaling activation is an efficient pathway for macrophage defense against bacterial invasion[16]. In cardiovascular system, ferroptosis contributes to ischemia- or drug-induced cardiomyopathy[17], acute renal failure[13], and atherosclerosis[18]. However, the role of ferroptosis in vascular senescence is understood poorly.

In the present study, we provide the first evidence that the pro-ferroptosis signaling in VSMCs is comprehensively activated under senescent condition, and blockade of ferroptosis signaling either pharmacologically or genetically, is able to alleviate NAD⁺ loss, stiffness, and senescence in vasculature. VSMC-specific overexpression of GPX4, but not catalytically inactive mutant GPX4, ameliorates experimental senescence and natural aging in mice vasculature. Furthermore, in a mouse model of abdominal aortic aneurysm (AAA), a vascular disease associated with aging, inhibition of ferroptosis signaling retards vascular senescence and pathological changes. Interestingly, pro-ferroptosis signaling drives the vascular senescence via secretome-dependent and independent manners in VSMCs. Mechanistically, we identify a previously undescribed role of peroxisome proliferator-activated receptor-γ (PPARγ) in 'ferroptosis-senescence' signal transduction by regulating nuclear receptor coactivator 4 (NCOA4)-centered ferritinophagy. Finally, the activated ferroptotic stress correlates with arterial stiffness in a human proof-of-concept study.

## Results

### Pro-ferroptosis signaling is activated upon senescence and aging conditions

We first investigated the ferroptosis status in replicative senescent primary VSMCs isolated from human carotid artery without atherosclerosis. These VSMCs had increased senescence-associated β-galactosidase (SA-β-gal) activity (Fig. 1A), and upregulation of senescent molecular markers p16^INK4A and p21^WAF1 (Fig. 1B). While the intracellular levels of NAD⁺ level showed significant declines in the senescent VSMCs (Fig. 1C), the NAD⁺ consumers CD38 and PARP-1 were upregulated (Fig. 1D). The protein expressions of ferroptosis signaling molecules, including acyl-CoA synthetase long-chain family member 4 (ACSL4), arachidonate 15-lipoxygenase (ALOX15) and transferrin receptor 1 (TFR1), divalent metal transporter-1 (DMT1) and iron-regulatory protein-2 (IRP2), were higher in senescent VSMCs (Fig. 1E).

The malondialdehyde (MDA) and 4-hydroxynonenal (4-HNE)-adducted proteins, which are markers of lipid peroxidation[19], were also upregulated (Fig. 1E). Conversely, the anti-ferroptosis molecule GPX4 was downregulated (Fig. 1E). Pro-ferroptosis signaling in these senescent human VSMCs was further monitored by staining of C11-BODIPY dye, a ferroptosis-sensitive lipid peroxidation probe, as well as FerroOrange dye, a probe of intracellular ferrous iron pool. The fluorescence intensities of C11-BODIPY and FerroOrange were induced dramatically in replicative senescent VSMCs (Fig. 1F). The glutathione (GSH) content and GSH/GSSG ratio were significantly reduced (Fig. 1G).

Next, we examined the activation of pro-ferroptotic signaling in an experimental vascular senescence model by treating cultured VSMCs with combination of a hormone angiotensin II (Ang II) and bleomycin (Bleo). Ang II was routinely applied to induce vascular senescence in plenty of studies[20–22], including our previous works[23,24]. Bleomycin, a genotoxic agent with DNA-attacking ability, exerts potent activation on senescence in VSMCs[25–27]. Upon the combined pro-senescent stress under Ang II plus Bleo, SA-β-gal activity and phosphorylated level of nuclear H2A.X at Ser139 residue (γH2A.X), a molecular marker of senescence[28], were significantly enhanced (Supplementary Fig. 1A, B). Intracellular levels of NAD⁺ were lower (Supplementary Fig. 1C). Edu incorporation and cell viability assays showed the cell proliferation was inhibited in Ang II+Bleo-treated VSMCs (Supplementary Fig. 1D, E), while the ferrous iron level was induced in the Ang II+Bleo-treated VSMCs (Supplementary Fig. 1F). GSH/GSSG ratio was inhibited in the Ang II+Bleo-treated VSMCs (Supplementary Fig. 1G). Accordingly, GPX4 was downregulated while ACSL4, ALOX15, DMT1, IRP2, and TFR1 were upregulated in these cells (Supplementary Fig. 1H).

We further compared the pro-ferroptosis signaling in vasculature between aged mice (24-month-old) and young mice (2-month-old). Fluorescent immunohistochemistry clearly demonstrated the upregulation of ACSL4 and ALOX15, as well as the downregulation of GPX4 in aged mouse aortae (Fig. 1H). GSH/GSSG ratio were reduced by ~30% in the aged aortae without endothelial or perivascular adipose tissue (Fig. 1I). Similar changes were observed in NAD⁺ level (Fig. 1J), while NAD⁺ consumers (CD38 and PARP-1) were upregulated (Fig. 1K). Flow cytometry showed a significant increase in number of 7-AAD-positive dying cells in the single-cell suspensions extracted from arteries of aged mouse (Fig. 1L).

Additionally, we established a mouse model with experimental vascular senescence by infusion with Ang II (400 ng/kg/min) via ALZET Osmotic micropump subcutaneously as described in our previous studies[23,29]. A low dose of bleomycin (40 ng/kg/min) was administered via the micropump simultaneously, significantly lower (approximately 1%) than the dosage used to induce lung fibrosis through the subcutaneous route[30]. The aortae isolated from these mice showed remarkable increases in protein expression of p16^INK4A/p21^WAF1 and number of 7-AAD⁺ dying cells (Supplementary Fig. 2A, B), as well as decline in NAD⁺ level (Supplementary Fig. 2C). Immunohistochemistry (ACSL4 and GPX4), immunoblotting (ACSL4, ALOX15, DMT1, IRP2, and GPX4), and GSG/GSSG ratio assay confirmed the activation of ferroptosis signaling in the Ang II+Bleo-treated mice aortae (Supplementary Fig. 2D–G). Administration of Ang II+Bleo did not cause significant fibrosis in mouse lung and kidney (Supplementary Fig. 2H, I).

Moreover, Ang II+Bleo infusion in aged mice aggravated vascular pro-ferroptotic signaling marker ACSL4 and GSH contents (Supplementary Fig. 3A, B). Ang II+Bleo infusion in aged mice also induced the expression of senescence marker γH2A.X, loss of aortic NADf⁺, and upregulation of NAD⁺ consumers (Supplementary Fig. 3C–E). All these data demonstrate that pro-ferroptosis signaling is activated in vasculature under senescence and aging conditions, and may be the cause of vascular aging.

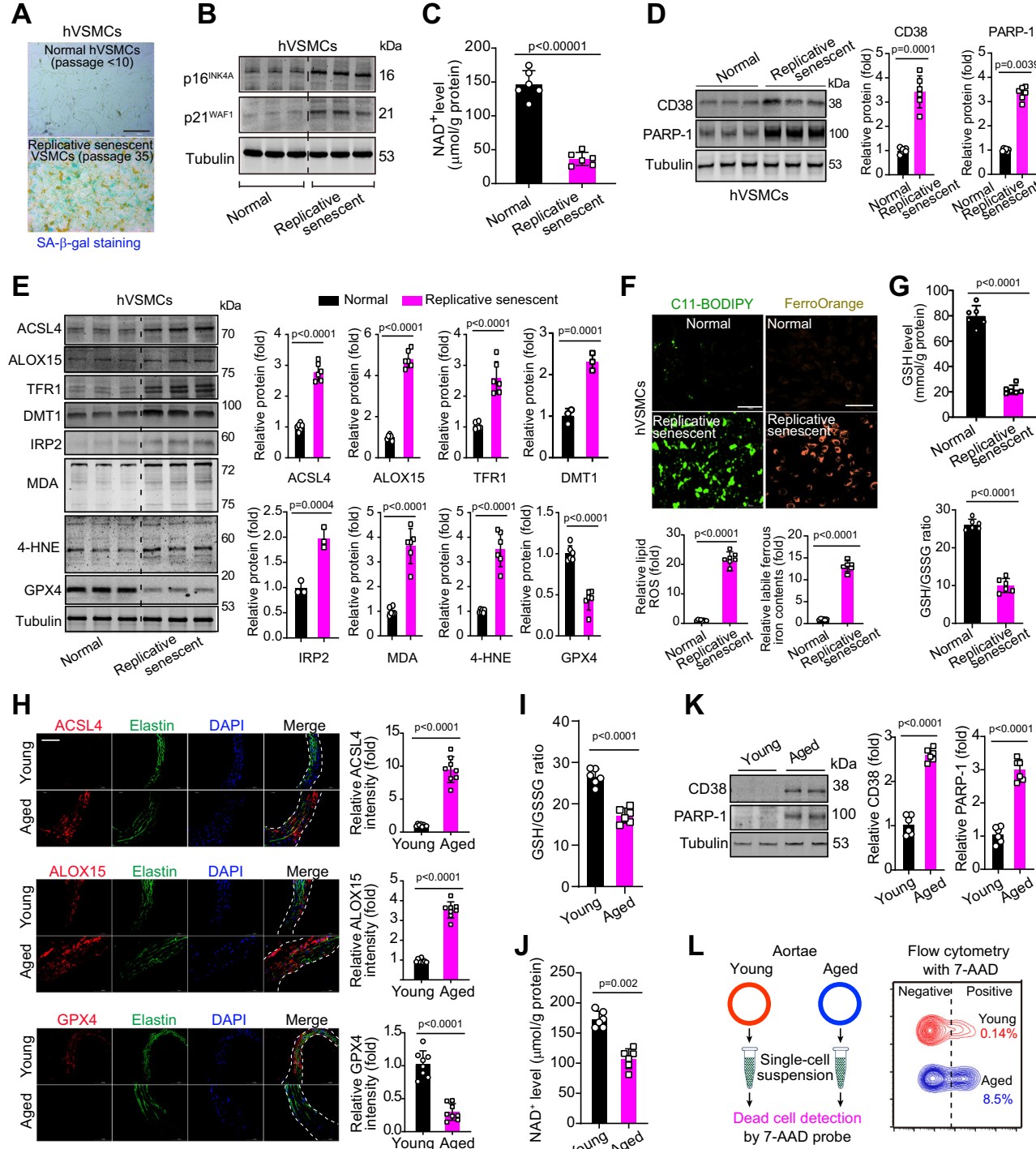

## Pharmacological inhibition of pro-ferroptosis signaling attenuates vascular NAD⁺ loss, senescence, and stiffness

To determine the causal relationship between pro-ferroptosis signaling and senescence, we treated senescent VSMCs with liproxstatin-1, a chemical inhibitor of ferroptosis[13]. Liproxstatin-1 administration not only pronouncedly reduced SA-β-gal activity and γH2A.X level (Supplementary Fig. 4A, B), but also ameliorated the inhibitory effects of Ang II+Bleo on EdU incorporation and cell viability (Supplementary Fig. 4C, D). Liproxstatin-1 reduced senescence markers (p53, p16$^{INK4A}$, and p21$^{WAF1}$), as well as inhibited the release of SASP factors interleukin-1α (IL-1α) and high mobility group box-1 protein (HMGB1) into culture medium from senescent VSMCs (Supplementary Fig. 4E, F). Intracellular NAD⁺ levels were depleted in senescent VSMCs but partially

rescued by liproxstatin-1 (Supplementary Fig. 4G). Corresponding changes in NAD⁺ consumers PARP-1 and CD38 were observed (Supplementary Fig. 4H).

We next tested the effects of ferroptosis inhibitor on senescence in mice. Liproxstatin-1 treatment suppressed SA-β-gal activity (Fig. 2A). Fluorescent immunohistochemistry analysis showed that p16$^{INK4A}$ and p21$^{WAF1}$ were scarcely expressed in normal aortae but abundantly expressed in Ang II+Bleo-infused aortic wall, which was partially attenuated by liproxstatin-1 (Fig. 2B, C). Similar inhibitory effects of liproxstatin-1 on HMGB1 and senescent markers (γH2A.X and IL-1α) was observed in the aortae (Fig. 2D, E). The declines of NAD⁺ contents, as well as the induction of NAD⁺ consumers PARP-1 and CD38 in aortae tissue by Ang II+Bleo infusion, were reversed by liproxstatin-1 (Fig. 2F,

**Fig. 1 | Pro-ferroptosis signaling is activated upon pro-senescence and aging conditions. A** Senescence-associated-β-galactosidase (SA-β-gal) staining was applied to evaluate senescence in primary cultured normal (passage 2-10) and replicative senescent (passage > 35) human VSMCs. Scale bar, 100 μm. The experiments were repeated three times, and the representative images were presented. **B** Representative immunoblotting images of p16[INK4A] and p21[WAF1]. **C** The intracellular NAD$^+$ level. $n = 6$–8 biologically independent samples. $n = 6$ biologically independent samples. **D** Representative immunoblotting and quantitative analyses of NAD$^+$ consumers CD38 and PARP-1. $n = 6$ biologically independent samples. **E** Representative immunoblotting and quantitative analyses of pro-ferroptosis factors (ACSL4, ALOX15, TFR1, DMT1, IRP2, MDA, and 4-HNE) and anti-ferroptosis factor (GPX4). $n = 3$ (for DMT1 and IRP2) or 6 (for ACSL4, ALOX15, TFR1, MDA, 4-HNE, and GPX4) biologically independent samples. **F** Lipid peroxidation and intracellular ferrous iron pool were monitored using C11-BODIPY probe and FerroOrange probe respectively in human primary VSMCs. Scale bar, 100 μm. $n = 6$ biologically independent samples. **G** The GSH contents and GSH/GSSG ratio in

human primary VSMCs. $n = 6$ biologically independent samples. **H** Representative immunohistochemistry staining and quantitative analyses of pro-ferroptosis factors (ACSL4 and ALOX15) and anti-ferroptosis factor (GPX4) in mice aortae. Elastin was stained by anti-elastin and nuclei was stained by DAPI. Scale bar, 100 μm. $n = 8$ biologically independent samples. **I** GSH/GSSG ratio in mice aortae. $n = 6$ biologically independent samples. **J** Intracellular NAD$^+$ content in mice aortae. $n = 6$ biologically independent samples. **K** Representative immunoblotting and quantitative analyses of NAD$^+$ consumers CD38 and PARP-1 in aortae from young and aged mice. $n = 6$ biologically independent samples. **L** Flow cytometer analysis of dead cells with 7-AAD probe in aortae from young and aged mice. Data expressed as mean ± SEM. Comparisons of parameters were performed with unpaired two-sided Student $t$-test. 4-HNE 4-hydroxynonenal, GPX4 glutathione peroxidase 4, ACSL4 acyl-CoA synthetase long-chain family member 4, ALOX15 arachidonate 15-lipoxygenase, TFR1 transferrin receptor, DMT1 divalent metal transporter-1, IRP2 iron-regulatory protein-2, MDA malondialdehyde, hVSMCs human vascular smooth muscle cells. Source data are provided as a Source Data file.

G). Pulse wave velocity (PWV), a golden standard for assessing arterial stiffness and vascular aging[31], was monitored in mice aortae. We found PWV was largely enhanced by Ang II+Bleo infusion but partially attenuated by liproxstatin-1 (Fig. 2H). The induced systolic blood pressure (SBP) and pulse pressure, another index of arterial stiffness[32], was also slightly but significantly suppressed by liproxstatin-1 (Fig. 2I). We next determined elastin contents. Elastin and elastic fibers, produced during childhood by adjacent VSMCs for smoothening the blood flow and pressure generated by the heart, are progressively degraded by mechanical stress and enzymatic activities in adulthood and aging, partly because of VSMCs senescence and death[2]. Elastica-van-Gieson (EVG) staining showed the elastin degradation in Ang II+Bleo-infused mice with was inhibited by liproxstatin-1 (Fig. 2J). Accordingly, the increased collagen content in Ang II+Bleo-infused mice was also reduced by liproxstatin-1 (Fig. 2K). All these indicate that pharmacologically inhibition of ferroptosis signaling activation blocks vascular NAD$^+$ loss, remodeling, and stiffness.

### Genetic inhibition of pro-ferroptosis signaling ameliorates vascular NAD$^+$ loss, senescence, and stiffness

To exclude the possible nonspecific effect of chemical inhibitor, we next used a mouse strain with genetic knockin of CAG-promoter-driven GPX4 cDNA into the Rosa26 locus to increase GPX4 protein level globally as described in our previous work (Supplementary Fig. 5A–D)[33]. This *Rosa26-GPX4-knockin* mouse stain was referred as 'R26-GPX4' hereafter. In R26-GPX4 mice aorta, the mRNA and protein expressions of GPX4 were substantially upregulated (Supplementary Fig. 5E, F). Arterial morphologies were comparable, and ferroptosis signaling activation or senescence was barely detected in the young R26-GPX4 mice and their wild-type (WT) control (data not shown). However, when the WT and R26-GPX4 mice were infused with Ang II+Bleo via Osmotic pump for 2 weeks to induce vascular senescence, the R26-GPX4 mice showed inhibited ferroptosis markers and rescued GSH/GSSG ratio in aortae (Supplementary Fig. 6A-B). Moreover, the senescence markers (γH2A.X, p53, p21[WAF1], and p16[INK4A]) in aortae of R26-GPX4 mice were significantly inhibited (Fig. 3A). The decline of NAD$^+$ in aortae were partially reduced, and the increases of NAD$^+$ consumers PARP-1 and CD38 protein expressions in aortae were attenuated in R26-GPX4 mice (Fig. 3B, C). Importantly, R26-GPX4 mice displayed significant inhibited PWV and PP after Ang II+Bleo infusion (Fig. 3D). These suggest knockin of GPX4 alleviates experimental vascular senescence in adult mice.

We next assessed whether genetic inhibition of ferroptosis could postpone the natural aging process of blood vessels. The ferroptosis markers and GSH/GSSG ratio were reduced and increased respectively in aortae of aged R26-GPX4 mice (24-month-old) compared with those from aged WT mice (Supplementary Fig. 6C, D). Fluorescent

immunohistochemistry confirmed the reduced senescent markers (γH2A.X and HMGB1) in aortic wall of aged R26-GPX4 mice compared with that of WT mice (Fig. 3E). The protein expression of p21[WAF1] and p16[INK4A] were decreased in aortae of aged R26-GPX4 mice (Fig. 3F). The mRNA levels of SASP factors, including *Tnf-α, Il-8, Il-6*, and *Il-1α*, were significantly lower in aged R26-GPX4 mice aortae (Fig. 3G). NAD$^+$ levels and were higher, while PARP-1 and CD38 were reduced in aged R26-GPX4 mice aortae (Figs. 3H, I). Histological staining analyses of EVG, Masson's trichrome, and Alizarin Red clearly demonstrated the reduced elastin degradation, collagen fibers, and calcification in aged R26-GPX4 mice respectively (Fig. 3J). The PWV and PP of aged R26-GPX4 mice aortae were lower than that in aged WT mice (Fig. 3K, L).

We also confirmed the influences of GPX4 overexpression on senescence in VSMCs. Enforced overexpression of GPX4 in murine aortic vascular smooth muscle cell line (MOVAS) resulted in a decrease in SA-β-gal activity, inhibited expression of p21[WAF1], p16[INK4A], and γH2A.X, as well as increases in EdU incorporation and cell viability (Supplementary Fig. 7A–E). GPX4 overexpression reversed the decline of NAD$^+$ and inhibited protein expression of PARP-1 and CD38 (Supplementary Fig. 7F, G). Taken together, all these in vivo and in vitro results indicate that genetic inhibition of ferroptosis signaling activation ameliorates vascular NAD$^+$ loss, senescence, and stiffness.

### VSMC-specific overexpression of GPX4, but not catalytically inactive mutant GPX4$^{Cys}$, ameliorates vascular senescence or aging in mice

It is known that GPX4 is a selenocysteine-containing protein, and mutation of selenocysteine to cysteine almost totally diminishes its anti-ferroptosis activity[34]. To show that smooth muscle ferroptosis plays a dominant role in vascular senescence/aging, we constructed two adeno-associated virus serotype-2/9 (AAV) encoding mouse WT GPX4 (GPX4$^{wt}$) or mutant GPX4$^{Cys}$ (Fig. 4A) under the control of an VSMC-specific SM22a promoter (AAV2/9-ZsGreen1-SM22a-GPX4$^{wt}$ and AAV2/9-ZsGreen1-SM22a-GPX4$^{Cys}$) and their control virus (AAV2/9-ZsGreen1-CTRL). We injected these AAV particles into 8-week-old mice for two times via tail vein and infused these mice with Ang II+Bleo (Supplementary Fig. 8A). VSMC-specific gene delivery by these AAV was confirmed using analyses of immunofluorescence (Supplementary Fig. 8B) and immunoblotting (Supplementary Fig. 8C). VSMC-specific overexpression of GPX4$^{wt}$ blocked the Ang II+Bleo-induced alterations in GSH/GSSG ratio (Fig. 4B), ACSL4 and TFR1 protein expression (Supplementary Fig. 8D), and MDA content (Supplementary Fig. 8E) in arterial tissue, confirming its inhibitory effect on pro-ferroptosis signaling. Immunohistochemistry showed VSMC-specific overexpression of GPX4$^{wt}$ inhibited p16[INK4A] accumulation in arterial wall (Fig. 4C). Immunoblotting showed similar changes in p21 and γH2A.X (Fig. 4D). Moreover, VSMC-specific overexpression of GPX4$^{wt}$ restored NAD$^+$

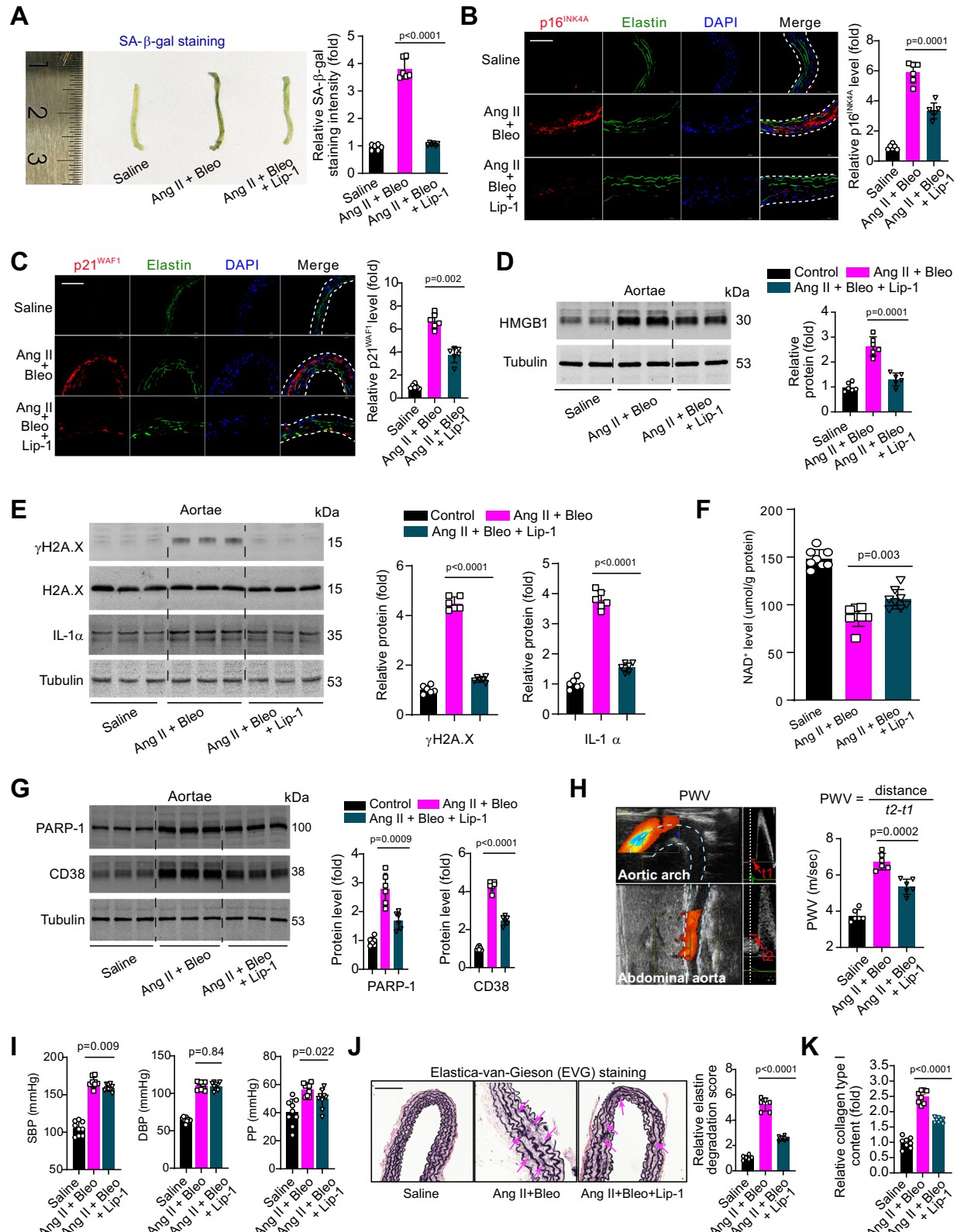

content (Fig. 4E) and arterial stiffness (PWV and PP, Fig. 4F). In contrast, administration of AAV carrying mutant GPX4$^{Cys}$ failed to produce these beneficial effects (Fig. 4B–F).

We next explored whether the VSMC-specific overexpression of GPX4 can delay vascular aging in natural condition. In 18-month-old mice, AAVs were injected into tail vein every 2 months after the first injection and total for 3 times; the aortae of these mice were examined at 24-month-old (Supplementary Fig. 8F). AAV2/9-ZsGreen1-SM22a-GPX4$^{wt}$, but not AAV2/9-ZsGreen1-SM22a-GPX4$^{Cys}$, showed significant inhibitory actions on senescence, evidenced by the results from GSH/GSSG ratio (Fig. 4G), ACSL4 protein expression (Fig. 4H), MDA content (Fig. 4I), and expressions of p16$^{INK4A}$ (Fig. 4J), p21 and γH2A.X (Fig. 4K). Additionally, AAV2/9-ZsGreen1-SM22a-GPX4$^{wt}$, but not AAV2/9-ZsGreen1-SM22a-GPX4$^{Cys}$, increased NAD$^+$ contents (Fig. 4L), as well as

**Fig. 2 | Ferroptosis inhibitor liproxstain-1 attenuates vascular NAD+ loss, senescence, and stiffness. A** Representative senescence-associated-β-galactosidase (SA-β-gal) staining and quantitative analysis in aortae from three groups of mice, including control group, angiotensin II (Ang II) + bleomycin (Bleo)-infused group and Ang II + Bleo + liproxstain-1 (Lip-1)-infused group. The combined treatment of Ang II+Bleo was used to induce vascular senescence. $n = 6$ biologically independent samples. **B, C** Representative fluorescent immunohistochemistry images and quantitative analyses of p16INK4A (**B**) and p21WAF1 (**C**) in aortae from three groups of mice. Scale bar, 100 mm. Elastin was stained by anti-elastin and nuclei was stained by DAPI. $n = 6$ biologically independent samples. **D** Immunoblotting analysis of high mobility group box-1 (HMGB1) in aortae from three groups of mice. $n = 6$ biologically independent samples. **E** Immunoblotting analysis of γH2A.X and IL-1α in aortae from three groups of mice. $n = 6$ biologically independent samples. **F** Intracellular levels of NAD+ in aortic tissue from three groups of mice. $n = 8$ biologically independent samples. **G** Immunoblotting analysis of two NAD+ consumers poly-ADP-ribose polymerase-1 (PARP-1) and CD38 in aortae from three groups of mice. $n = 6$ biologically independent samples. **H** Arterial stiffness was evaluated by pulse wave velocity (PWV) with a high-resolution animal ultrasound Vevo 2100 system. Ultrasound images of pulse waves at the aortic arch and the abdominal aorta were recorded and PWV was then calculated as showed in the figure. $n = 6$ biologically independent samples. **I** Systolic blood pressure (SBP), diastolic BP, and pulse pressure (PP) in the three groups of mice. $n = 8$ biologically independent samples. **J** Elastica-van-Gieson (EVG) staining was used to evaluate elastin degradation in the three groups of mice. $n = 6$ biologically independent samples. **K** Collagen content in the vasculature of mice. $n = 8$ biologically independent samples. Data expressed as mean ± SEM. Comparisons of parameters were performed with unpaired two-sided Student $t$-test between Ang II+Bleo and Ang II +Bleo+Lip-1 groups. The data in Saline group were not included in comparison. NS no significance. Source data are provided as a Source Data file.

reduced PARP-1 and CD38 expression (Fig. 4M). The PWV and PP were suppressed by AAV2/9-ZsGreen1-SM22a-GPX4wt but not by AAV2/9-ZsGreen1-SM22a-GPX4Cys (Fig. 4N). These results support the notion that the pro-ferroptosis signaling in smooth muscle cells governs vascular NAD+ homeostasis, senescence, and stiffness.

## Genetic inhibition of ferroptosis signaling mitigates vascular senescence and retards progression of abdominal aortic aneurysm (AAA)

To assess the pathological relevance of ferroptosis signaling in specific vascular disease state related to senescence[35–37], the WT and R26-GPX4 mice were subjected to a well-established mouse model of AAA induced by porcine pancreatic elastase perfusion (PPE model). SA-β-gal staining clearly demonstrated that the senescent cells accumulated in the inflated lesion site in abdominal aorta, and the SA-β-gal-positive area in R26-GPX4 mice was significantly lower than that in WT mice (Supplementary Fig. 9A, pink arrow). The protein expression of γH2A.X in AAA tissue of R26-GPX4 mice was less than that in WT mice (Supplementary Fig. 9B). GSH/GSSG ratio was increased in AAA tissue of R26-GPX4 mice (Supplementary Fig. 9C). The maximal AAA diameter in R26-GPX4 mice was lower compared with that in WT mice (Supplementary Fig. 9D). NAD+ content was enhanced in R26-GPX4 mice than WT mice (Supplementary Fig. 9E). Immunohistochemistry analysis demonstrated that the senescence-associated molecule HMGB1 was predominately expressed in tunica media and remarkably decreased in R26-GPX4 mice (Supplementary Fig. 9F). EVG staining showed that the elastic fibers degradation in AAA tissue was significantly alleviated in R26-GPX4 mice (Supplementary Fig. 9G). These results showed that genetic inhibition of ferroptosis signaling ameliorates vascular senescence and AAA pathology.

## Pro-ferroptosis signaling drives vascular senescence via secretome-dependent and independent manners in VSMCs

To consolidate the notion that pro-ferroptotic signaling is capable of induce senescence, we administrated a ferroptosis activator RSL3[11] in VSMCs. Administration of RSL3 at a non-cytotoxic dose (0.5 μM) demonstrated a rapid inhibitory effect (<24 h) on the GSH/GSSG ratio, persisting until Day 5 (Fig. 5A). However, at this dose, RSL3 did not induce significant cell death within the first 24 h of incubation. Inhibition of EdU incorporation and a decrease in cell viability became apparent at 3 days post-incubation (Fig. 5B, C). These findings suggest that the ferroptosis inducer RSL3 promptly induced pro-ferroptosis signaling, appearing to precede its effects on cell proliferation and death. At 5 days post-incubation, RSL3 at a non-cytotoxic dose (0.5 μM) significantly increased SA-β-gal activity (Fig. 5D), and upregulated p16INK4A and p21WAF1 (Fig. 5E), suggesting an induction of senescence by pro-ferroptosis signaling. To visualize the co-existence of senescence and pro-ferroptosis signaling within a cell, we employed a combination of the combination of 'ferroptosis probe' (Liperfluo probe[15], green) and

'senescence probe' (KSAP1, red)[38] to stain VSMCs treated with RSL3 at a non-cytotoxic dose (0.5 μM) for 3 days. The green signal (Liperfluo dye) was observed in almost all VSMCs, confirming the activation of pro-ferroptosis signaling in these cells (Fig. 5F). In contrast, the red signal from the senescence probe (KSAP1) was only present in some VSMCs (indicated by pink arrows) and not in all VSMCs. Interestingly, the 'non-senescent' VSMCs with little red signal (indicated by white arrows) maintained a spindle-shaped appearance, while the 'senescent' VSMCs with a red signal exhibited a flat and enlarged morphology, characteristic of recognized senescent VSMCs[39]. These results indicate the co-existence of pro-ferroptosis signaling and senescence within the same cells and suggest that the activation of the ferroptosis signaling may precede senescence in RSL3-treated VSMCs.

We next explored whether the cells undergoing ferroptosis would release factors to promote senescence in neighboring cells. Cultured VSMCs were treated with RSL3 at a cytotoxic dose (10 μM) and another ferroptosis activator erastin[11] for 24 h to induce ferroptosis, and then the medium containing RSL3 and erastin were removed. The VSMCs underwent ferroptosis were cultured for additional 24 h, and the culture medium was collected and transferred to medium of healthy VSMCs for another 5-day culture (Fig. 5G). We found both the transferred culture medium from cytotoxic dose of RSL3 and erastin-treated VSMCs resulted in significant increases in p16INK4A/p21WAF1 (Fig. 5H) and SA-β-gal activity (Fig. 5I), as well as decreases in EdU incorporation and cell viability (Supplementary Fig. 10A, B). These results suggest that ferroptotic cells can induce senescence in a secretome-dependent manner. We also evaluated whether the secretome of VSMCs undergoing other types of cell death such as apoptosis and necroptosis could also induce senescence. By contrast, the transferred culture medium from VSMCs underwent apoptosis (induced by ABT-737) and necroptosis (induced by TNF-α, SM-164 and z-VAD-fmk, referred as TSZ) failed to increase SA-β-gal activity (Fig. 5J). We next studied whether senescent cells would release SASP-related factors to promote ferroptosis (Fig. 5K). The culture medium derived from replicative senescent VSMCs did not cause any significant increase of MDA or decline of GSH/GSSG ratio (Fig. 5L). C11-BODIPY probe staining also showed that the culture medium derived from replicative senescent VSMCs failed to trigger ferroptotic process, whereas RSL3 evidently induced ferroptosis (Fig. 5M). Moreover, we evaluated the therapeutic efficacy of ROS scavenger N-acetyl-L-cysteine (NAC) and NAD+ precursor nicotinamide mononucleotide (NMN). Simultaneous treatment of NAC or NMN was able to partially block the senescence induced by Ang II+Bleo and ferroptosis inducer RSL3 (Supplementary Fig. 10C, D). However, delayed administration of NAC or NMN, was unable to reverse the already-formed senescence driven by Ang II+Bleo (Supplementary Fig. 10E, F). These results indicate that pro-ferroptosis signaling drives senescence via secretome-dependent and independent manners in VSMCs; however, senescent VSMCs seems to be unable to induce ferroptosis.

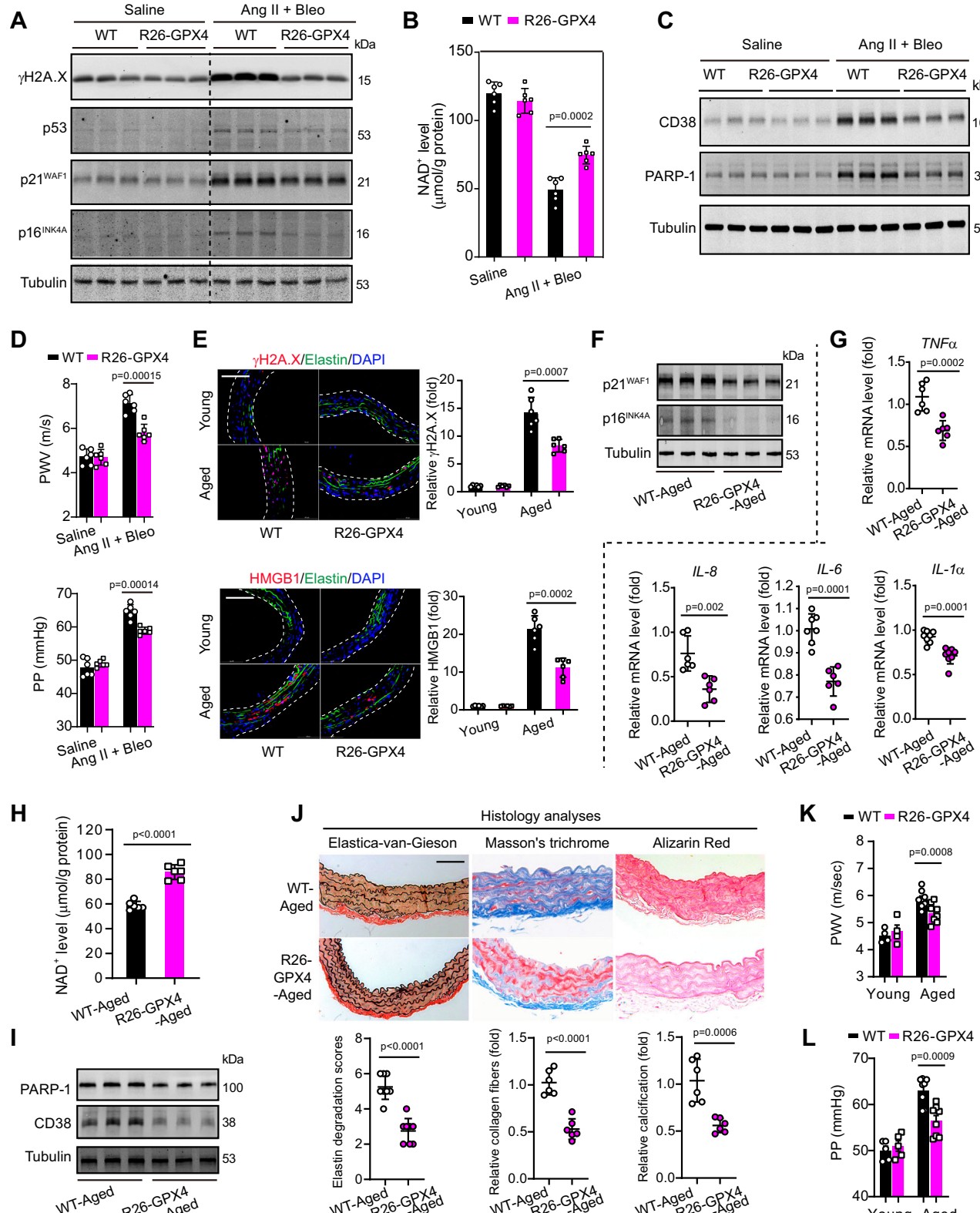

## Inhibition of ferroptotic signaling suppresses ferritinophagy with regulation on PPARγ signaling pathway

Ferritinophagy, a special form of autophagy associated with breakdown of ferritin and subsequent release of iron, critically contributes to the ferroptosis process in cytoplasm (Fig. 6A)[40]. In the ferritinophagy, the nuclear receptor coactivator 4 (NCOA4), which is previously thought as a transcriptional coactivator of nuclear androgen receptor-70 (ARA70)[41], has been recently found to acts as a cytosol ferritinophagic cargo receptor to deliver ferritin to lysosomes[40] via forming insoluble condensates[42,43]. Thus, we compared the ferritinophagic status in the full lysates of aortae between Ang II+Bleo-infused WT and R26-GPX4 mice. It was observed that the autophagy molecular marker LC3-II was significantly increased in aortae from WT mice and, to a much lesser extent, in R26-GPX4 mice (Fig. 6B). Moreover, NCOA4 and the ferritinophagy substrate ferritin heavy chain 1 (FTH1), were significantly upregulated and downregulated in WT mice aortae

**Fig. 3 | GPX4 knockin mice display ameliorated vascular NAD$^+$ loss, senescence, and stiffness. A** Immunoblotting analyses of senescence markers including γH2A.X, p53, p16$^{INK4A}$, and p21$^{WAF1}$ in aortae. The combined treatment of Ang II+Bleo was used to induce vascular senescence. **B** NAD$^+$ levels in aortic tissue of adult WT and R26-GPX4 mice. $n$ = 6 biologically independent samples. **C** Immunoblotting analyses of NAD$^+$ consumers PARP-1 and CD38 in aortic tissue. **D** Arterial stiffness was evaluated by pulse wave velocity (PWV) and pulse pressure (PP) values of adult WT and R26-GPX4 mice. $n$ = 6 biologically independent samples. **E** Representative fluorescent immunohistochemistry and quantitative analyses of senescence markers γH2A.X and HMGB1 in aortae tissue of young (8-week-old) or aged (24-month-old) WT and R26-GPX4 mice. Elastin was stained by anti-elastin and nuclei was stained by DAPI. $n$ = 6 biologically independent samples. Scale bar, 100 μm. **F** Immunoblotting analyses of senescence markers p16$^{INK4A}$ and p21$^{WAF1}$ in aortic

tissue of mice. **G** Gene expression of SASP factors (*Tnfα, Il-8, Il-6,* and *Il-1α*) in aortic tissue of mice. $n$ = 6 (*Tnfα, Il-8,* and *Il-6*) or 8 (*Il-1α*) biologically independent samples. **H** NAD$^+$ levels in aortic tissue of mice. $n$ = 6 biologically independent samples. **I** Immunoblotting analyses of NAD$^+$ consumers PARP-1 and CD38 in aortic tissue of mice. **J** Evaluation of elastin degradation, collagen fibers, and calcification by histological analyses with Elastica-van-Gieson, Masson's trichrome, and Alizarin Red staining respectively in aortae of mice. $n$ = 8 (elastin degradation) or 6 (collagen fibers and calcification) biologically independent samples. Scale bar, 50 μm. **K** PWV values of young and aged mice. $n$ = 4 (young mice) or 8 (aged mice). **L** PP values of young and aged mice. $n$ = 5 (young mice) or 8 (aged mice). Data expressed as mean ± SEM. Comparisons of parameters were performed with unpaired two-sided Student's *t*-test. The values in saline-infused mice or young mice were not used for statistical comparison. Source data are provided as a Source Data file.

respectively and, to much lesser extents in R26-GPX4 mice aortae (Fig. 6B). Similar changes were observed in aged WT and R26-GPX4 mice aortae (Fig. 6C). Moreover, in the VSMCs isolated from WT and R26-GPX4 mice, we evaluated the autophagy flux according to the guideline[44] in the presence of bafilomycin A1, and observed consistent changes (Supplementary Fig. 11A). Fluorescent immunohistochemistry showed that NCOA4 was barely expressed in the aortic tunica media of both young WT and R26-GPX4 mice; however, NCOA4 was abundantly expressed in aortic tunica media of aged mice, which was reduced by GPX4 overexpression (Fig. 6D). Electron microscopy was applied to determine the number of double-membrane autophagosomes, which reflects the ferritinophagic status. We observed that the number of autophagosome-like structures (red arrow) was reduced in aortae of aged R26-GPX4 mice than that of aged WT mice (Fig. 6E). These results suggest that inhibition of pro-ferroptotic signaling by GPX4 knockin blocks ferritinophagy in vasculature.

To identify how pro-ferroptosis signaling affects ferritinophagy upon pro-senescent stress, we performed a transcriptomics study using bulk RNA-sequencing in aortae (without tunica intima, tunica adventitia and perivascular fat) of WT and R26-GPX4 mice infused by Ang II+Bleo (Fig. 6F). Based on 2 selection criteria (fold change >2 and false discovery rate <0.01), 242 genes (56 genes were upregulated, and 186 genes downregulated) were identified as transcriptionally altered by GPX4 knockin in aorta tissue (Fig. 6G and Supplementary Table 1). Kyoto Encyclopedia of Genes and Genomes (KEGG) analysis was conducted, and very surprisingly we found the differentially expressed genes between the WT and R26-GPX4 aortae upon Ang II+Bleo were enriched in peroxisome proliferator-activated receptor (PPAR) signaling pathway (Fig. 6H), a nuclear receptor that is important in many physiological and pathological processes such as insulin sensitivity and lipid metabolism[45]. PPARs also exert anti-atherogenic and anti-inflammatory effects on the vascular wall[45]. In the most significantly differentially expressed genes, PPAR-related genes were obviously noted (Fig. 6I). Gene Set Enrichment Analysis (GSEA) also revealed that ferroptosis inhibition significant downregulated gene sets including fatty acid metabolic process and cellular lipid catabolic process, which were highly controlled by PPARs signaling pathway (Fig. 6J)[45].

We next explored the importance of PPAR signaling pathway in vascular senescent-associated ferritinophagy. Real-time PCR analysis confirmed the downregulation of several key genes in PPAR pathway, including *Ppara, PPARγ, PPARγc1a,* and *PPARγc1b* (Supplementary Fig. 11B). Next, we used FerroOrange probe combined with RNA interference to further identify which PPAR pathway-associated molecule may be the key player in ferritinophagy regulation. Unexpectedly, we found that knockdown of PPARγ significantly abolished the increase of ferrous iron in VSMCs upon Ang II+Bleo-induced senescence stress (Supplementary Fig. 11C). By contrary, depletion of PPARα (encoded by *Ppara*), PPARγ coactivator-1 α (PGC-1α, encoded by *PPARγc1a*) and PPARγ coactivator-1β (PGC-1β, encoded by *PPARγc1b*) failed to affect the increase of labile ferrous iron pool

(Supplementary Fig. 11C), suggesting that the nuclear function of PPARγ was not involved in its regulation on ferritinophagy. C11-BODIPY assay confirmed that PPARγ knockdown further aggravated senescence-associated lipid peroxidation (Fig. 6K). Conversely, activation of PPARγ using rosiglitazone reduced γH2AX staining (Fig. 6L). These results indicate that inhibition of ferroptosis signaling may suppress ferritinophagy and affect non-nuclear PPARγ signaling in vascular senescence.

## Cytoplasmic PPARγ interacts with NCOA4, leading to the dissociation of the NCOA4-ferritin ferritinophagic complex

As ferritin is a cytoplasmic iron storage protein and NCOA4-ferrtin ferritinophagic complex was observed in cytoplasm[40,46], we wondered how PPARγ affects ferritinophagy. As PGC-1α and PGC-1β always act as co-factors to interact with PPARs and thus form a protein complex in nucleus to regulate gene transcription, we proposed that the PPARγ itself, rather than the nuclear complex, is vital for the ferritinophagy activation upon Ang II+Bleo. As expected, immunoblotting analysis showed that the total PPARγ protein expression in aortic extract was unexpectedly unchanged by ferroptosis inhibition (Fig. 7A); however, the cytosol PPARγ was induced (Fig. 7B) whereas the nuclear PPARγ was reduced (Fig. 7C), suggesting that ferroptosis inhibition promotes nuclear-cytoplasmic shuttling of PPARγ[47]. This effect was confirmed through experiments utilizing the ferroptosis inhibitor liproxstatin-1 (Supplementary Fig. 12). To further answer the influence of inhibition of ferroptosis signaling and subsequent increase of cytoplasmic PPARγ on ferritinophagy, we overexpressed GPX4 or administrated PPARγ agonist rosiglitazone in MOVAS cell, and observed that the upregulation of NCOA4 upon Ang II+Bleo-induced senescence stress was blocked by PPARγ overexpression and PPARγ agonist rosiglitazone (Supplementary Fig. 13A). Accordingly, GPX4 overexpression also blunted the upregulation of NCOA4 by Ang II+Bleo (Supplementary Fig. 13B). In contrast, PPARγ antagonist T0070907 or knockdown of PPARγ reversed the inhibition of NCOA4 by GPX4 overexpression (Supplementary Fig. 13C). Further, similar to GPX4 overexpression, PPARγ overexpression or PPARγ agonist rosiglitazone alleviated Ang II +Bleo-induced senescence markers p16$^{INK4A}$ and p21$^{WAF1}$ (Supplementary Fig. 13D).

Interestingly, in a public microarray dataset (No. #GSE1011, Supplementary Fig. 14A) aimed to explore the altered genes in aorta of mice treated with PPARγ agonist rosiglitazone, we found PPARγ activation upregulated anti-ferroptotic genes including *Gpx4, Nfs1, Dhodh, Prdx6,* etc., and suppressed pro-ferroptotic genes including *Atf3* and *Dusp1* in mice aorta (Supplementary Fig. 14B). Conversely, two ferritinophagic factors *Ncoa4* and *Fth1* were downregulated by PPARγ activation in aorta (Supplementary Fig. 14B). In support of these, in this public dataset, *Gpx4* was negatively correlated ferritinophagic genes (*Ncoa4 and Fth1*) and pro-ferroptotic genes (*Atf3, Zeb1,* and *Dusp1*), but positively correlated with anti-ferroptosis gene such as *Dhodh, Nsf1 and Prdx6* (Supplementary Fig. 14C). These results indicate that

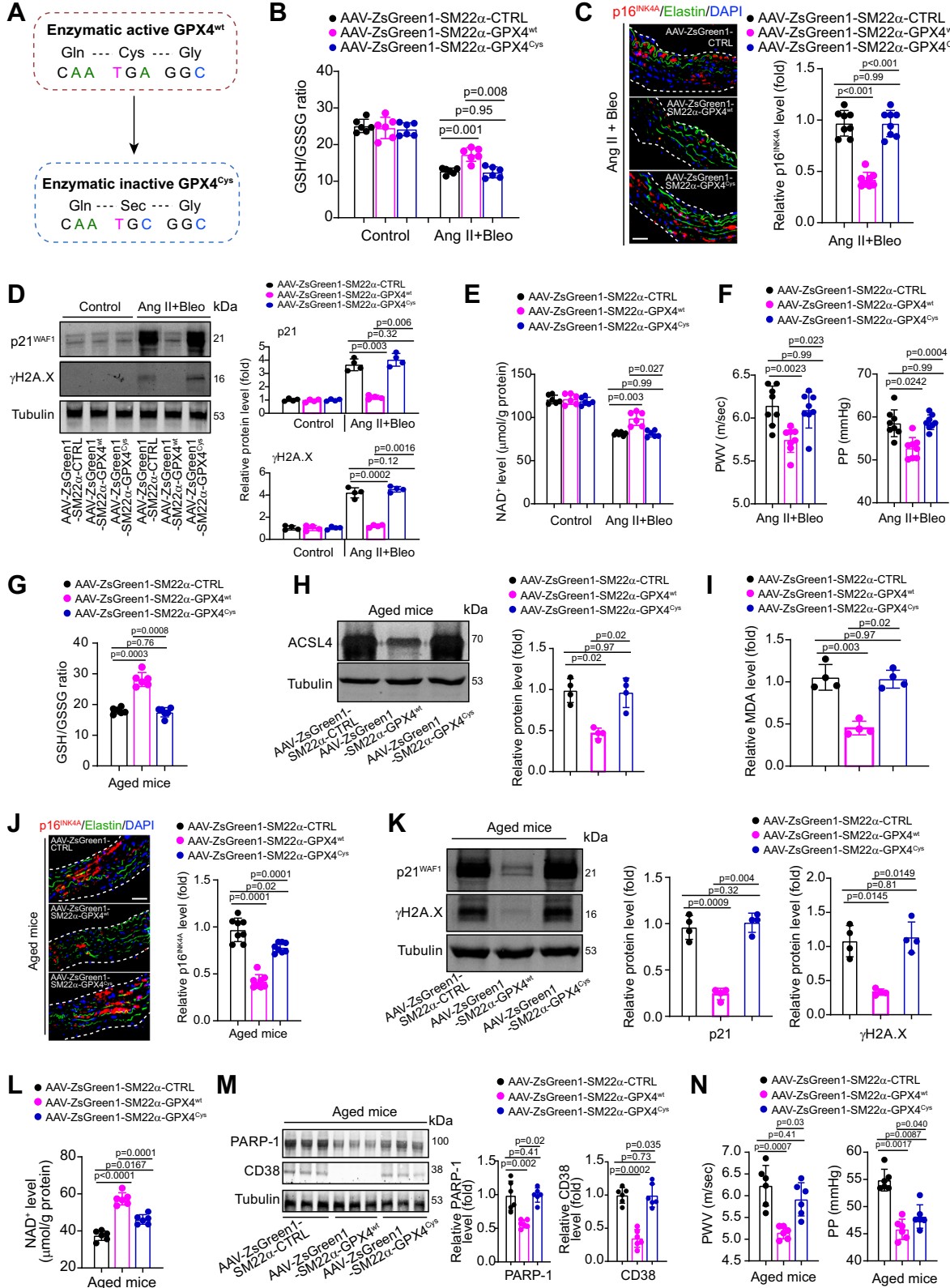

inhibition of pro-ferroptotic signaling promotes the inhibitory action of PPARγ on NCOA4-mediated ferritinophagy via facilitating its nuclear-cytoplasmic shuttling.

Based on the above results, we speculated that in response to stress, the elevated cytoplasmic PPARγ may compete with ferrtin in binding to NCOA4, and thus exerts a 'braking' effect on ferrtin degradation (ferritinophagy) and ferroptosis (Fig. 7D). To test this

speculation, we detected the interaction between PPARγ and NCOA4 in VSMCs. Indeed, such a direct interaction between PPARγ and NCOA4 was observed (Fig. 7E), which was in line with the previously reported association between PPARγ and NCOA4[48]. Interestingly, we observed that this interaction was further strengthened by PPARγ agonist rosiglitazone (Fig. 7E). Immunofluorescence showed that NCOA4 and PPARγ exhibited diffuse localization in cytoplasm and

**Fig. 4 | VSMC-specific overexpression of GPX4, but not catalytically inactive mutant GPX4$^{Cys}$, ameliorates vascular senescence, and aging in mice.**
**A** Construction of AAV containing the active site of GPX4 (GPX4$^{wt}$) or the catalytically inactive mutant GPX4$^{cysteine}$ (GPX4$^{cys}$). In GPX4$^{cys}$, the selenocysteine was mutated to cysteine. **B** GSH/GSSG ratio in aortae of mice. $n = 6$ biologically independent samples. **C** Representative fluorescent immunohistochemistry and quantitative analyses of senescence markers p16$^{INK4A}$ in aortae of mice. Elastin was stained by anti-elastin and nuclei was stained by DAPI. $n = 8$ biologically independent samples. Scale bar, 20 μm. **D** Immunoblotting analyses of p21 and γH2A.X in aortae of mice. $n = 4$ biologically independent samples. **E** NAD$^+$ contents in aortae of mice. $n = 6$ biologically independent samples. **F** Arterial stiffness parameters (PWV and PP) in mice. $n = 8$ biologically independent samples. **G** GSH/GSSG ratio in aortae of aged mice. $n = 6$ biologically independent samples. **H** Immunoblotting analysis of ferroptosis marker ACSL4 in aortae of aged mice. $n = 4$ biologically independent

samples. **I** MDA contents in aortae of aged mice. $n = 4$ biologically independent samples. **J** Representative fluorescent immunohistochemistry and quantitative analyses of senescence markers p16$^{INK4A}$ in aortae of aged mice. Elastin was stained by anti-elastin and nuclei was stained by DAPI. $n = 8$ biologically independent samples. Scale bar, 20 μm. **K** Immunoblotting analyses of p21 and γH2A.X in aortae of aged mice. $n = 4$ biologically independent samples. **L** NAD$^+$ level and GSH/GSSG ratio in aortae of aged mice. $n = 6$ biologically independent samples.
**M** Immunoblotting analyses of NAD$^+$ consumers PARP-1 and CD38 in aortae of aged mice. $n = 6$ (for PARP-1) or 7 (for CD38) biologically independent samples. **N** Arterial stiffness parameters (PWV and PP) in aged mice. $n = 6$ biologically independent samples. Data expressed as mean ± SEM. Comparisons of parameters were performed with One-Way ANOVA followed by a Tukey's multiple comparisons test. Source data are provided as a Source Data file.

nuclei respectively in normal VSMCs and there was no colocalization between these two proteins (Fig. 7F and Supplementary Fig. 15). Upon Ang II+Bleo stress, both NCOA4 and PPARγ formed dot-like structures in cytoplasm and some colocalization between these two factors was observed. When PPARγ was activated by rosiglitazone, PPARγ was highly co-localized with PPARγ in the puncta-like structures in cytoplasm (yellow arrows, Fig. 7F).

Next, we explored the influence of PPARγ activation on the interaction between FTH1 and NCOA4 in ferritinophagy. To clearly show the ferritinophagy, we generated a GFP-tagged NCOA4 plasmid, transfected it into VSMCs and treated the VSMCs with Ang II+Bleo. In control VSMCs upon stress, GFP-tagged NCOA4 (green) accumulated in puncta-like structures, which co-localized with lysosomes (LysoTracker, red), indicating a promoted ferritinophagy process by stress; however, in the rosiglitazone -treated VSMCs, GFP-tagged NCOA4 failed to form such structures (Fig. 7G), suggesting that the fusion of NCOA4 with lysosomes is impaired by PPARγ activation. In situ proximity ligation assay (PLA) showed that there was no positive dot (red) in normal condition; however, the positive dots appeared upon Ang II+Bleo treatment, suggesting that FTH1 and NCOA4 are in close proximity under this stress (Fig. 7H). Importantly, the PLA-positive dots were observed around the nuclei, suggesting that the interaction between FTH1 and NCOA4 occurs at the cytoplasmic region outside the nuclear membrane boundary. Treatment of ROSI significantly repressed the number of PLA-positive dots (Fig. 7H), suggesting PPARγ activation retards the interaction between FTH1 and NCOA4. Co-immunoprecipitation analysis of FTH1 and NCOA4 confirmed these results (Fig. 7I). In fact, NCOA4 has been intensively studied before the discovery of ferritinophagy in 2014[40]. NCOA4 has a LxxLL motif (PPARγ-binding site) which are illustrated in Fig. 7J[49]. We engineered a plasmid containing a mutant LxxLL motif within NCOA4 (ΔNCOA4$^{LxxAA}$), where the two leucines were substituted with alanines. Our findings indicated that compared with the interaction between PPARγ and NCOA4, the interaction between PPARγ and ΔNCOA4$^{LxxAA}$ was significantly inhibited, although not completely abolished (Fig. 7K). In support of this observation, overexpression of wild-type NCOA4 in NCOA4-depleted VSMCs rescued the activation of ferroptosis signaling induced Ang II+Bleo stress, as evidenced by the augmentation of the liable Fe$^{2+}$ pool and lipid peroxidation lipid peroxidation levels. Conversely, overexpression of ΔNCOA4$^{LxxAA}$ failed to elicit similar effects, underscoring the pivotal role of the LxxLL motif in the interaction between PPARγ and NCOA4 and its impact on ferroptosis signaling activation (Fig. 7L). Taken together, all these results suggest that PPARγ interacts with the LxxLL motif of NCOA4 to disassociate NCOA4-ferrtin ferritinophagic complex.

### Pro-ferroptosis signaling is associated with arterial aging/stiffness in human

Next, we examined the association of pro-ferroptosis signaling and arterial aging/stiffness in humans. Carotid artery tissue samples were collected from middle-aged (<45 years old) and elderly (>65 years old) individuals, taken from regions outside pathological lesions during carotid artery aneurysm resection (Fig. 8A). Immunohistochemistry analysis revealed significantly higher expression of pro-ferroptosis markers ACSL4 and ALOX15 in carotid arteries of elderly patients compared to middle-aged subjects (Fig. 8A, B). Elevated levels of Fe$^{2+}$ and MDA were observed in the carotid arteries of the elderly group (Fig. 8D, E), accompanied by lower GSH/GSSG ratio and NAD$^+$ levels (Fig. 8F, G).

In addition, we assessed three ferroptosis signaling-related molecules (4-HNE, MDA, and 15-hydroxyeicosatetraenoic acid [15-HETE]) in plasma from a small group of 73 individuals aged 47–90 years. Carotid-femoral pulse wave velocity (cfPWV) values were measured using SphygmoCor® technology as described previously[24]. In this population, we observed significant positive associations between plasma levels of 4-HNE, MDA, and 15-HETE, respectively, and PWV values (Fig. 8H–J). These results from our proof-of-concept study indicate a positive relationship between pro-ferroptosis signaling and vascular aging/stiffness in human.

## Discussion

Our findings reveal that activation of ferroptosis signaling, a novel iron-dependent lipid peroxidation-driven form of regulated cell death (RCD), in senescent VSMCs and the vascular wall of aged human and animal models. Pharmacological or genetic inhibition of ferroptosis signaling effectively mitigates these pathological alterations, slows down vascular aging-like phenotypes in vitro and in vivo, and provides protection against vascular aging-associated disorders such as abdominal aortic aneurysm (AAA). These effects are mediated through the regulation of NCOA4-ferritin ferritinophagy. A schematic representation of these findings is depicted in Fig. 8K.

This study used a combined method (Ang II plus bleomycin) to induce vascular senescence and aging. Ang II is one of the most commonly used agents to induce vascular senescence due to its various detrimental actions on vasculature[20–22]. Our group previously used this method to induce senescence in cultured VSMCs and animal aortae[23,24]. In the present study, we infused Ang II plus bleomycin simultaneously into mouse. Bleomycin is a genotoxic agent with DNA-attacking ability and exerts potent activation on senescence in VSMCs[25,26]. According to our results, the combination of Ang II and bleomycin swiftly induced VSMCs senescence and arterial stiffness in mice, evidenced by the SASP accumulation, elastic fibers degeneration, collagen deposit and PWV increment. Previously, PP was reported to predicts vascular stiffness[50]. In the past decade, PWV has emerged as a more sensitive method and has thought to be the gold standard noninvasive tool for measuring vascular stiffness in aged individuals[31]. We found inhibition of ferroptosis signaling activation lowered PWV and PP simultaneously, supporting the notion that pro-ferroptosis signaling promotes vascular stiffness. We noted that Ang II and bleomycin induce senescence across cardiovascular tissues via various

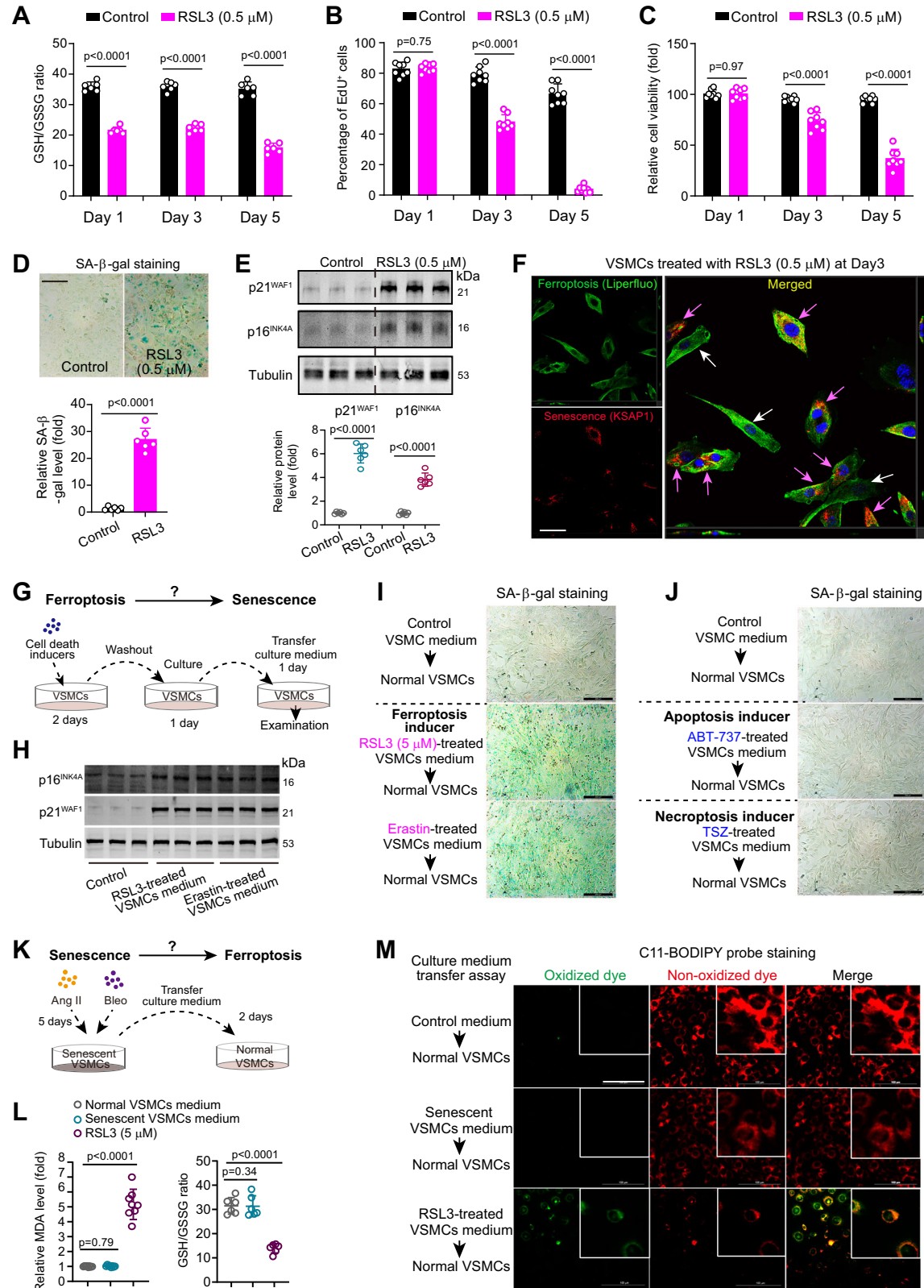

pathological mechanisms, including DNA damage, mitochondrial dysfunction, and SASP[20–22,25–27]. Thus, there might be more molecular events contributing to the senescence observed in our in vitro and in vivo model, which are independent of pro-ferroptotic signaling.

Our findings uncover an unknown link between pro-ferroptosis signaling and senescence in vasculature. Although senescence does not constitute a form of RCD, emerging evidence has shown that senescence is largely associated with RCD such as apoptosis[51] and autophagy[52]. Recently, a possible relationship between ferroptosis and senescence has been proposed. p53, one of the most important inducers of cell senescence, has been shown to regulate cystine metabolism and tumorigenic ferroptosis[53]. Iron was found to be accumulated in radiation-induced senescent mouse embryonic fibro-blasts with ferroptosis inhibition[54]. In retinal pigment epithelial cell,

**Fig. 5 | Pro-ferroptosis signaling drives vascular senescence via secretome-dependent and independent manners in VSMCs. A–C** GSH/GSSG ratio, percentage of EdU[+] incorporation, and relative cell viability of VSMCs treated with PBS (control) or RSL3 at a non-cytotoxic dose (0.5 μM) for 1, 3, and 5 days. $n = 8$ (**A**) or 6 (**B**) biologically independent samples. **D** Representative images and quantitative analyses of SA-β-gal activity in mouse VSMCs treated as in (**A–C**) for 5 days. $n = 6$ biologically independent samples. Scale bar, 100 μm. **E** Immunoblotting analysis of p16[INK4A] and p21[WAF1] in mouse VSMCs. $n = 6$ biologically independent samples. **F** Representative images for VSMCs treated by RSL3 at a non-cytotoxic dose (0.5 μM) for 3 days and then incubated with a ferroptosis probe (Liperfluo, green signal) plus senescence probe (KSAP1, red signal) for 1 h. Scale bar, 20 μm. **G** A schematic representation of study design to determine whether ferroptotic cell medium induces senescence. RSL3 at a cytotoxic dose (10 μM) and another ferroptosis activator erastin at a cytotoxic dose (20 μM) were used. **H** Representative images immunoblotting analysis of p16[INK4A] and p21[WAF1] in mouse VSMCs cultured with the transferred medium. $n = 3$ biologically independent samples.

**I** Representative images of SA-β-gal staining for VSMCs cultured with the transferred medium from RSL3- or erastin-treated VSMCs respectively. **J** Representative images of SA-β-gal staining for VSMCs cultured with the transferred medium from apoptotic VSMCs (ABT-737-treated, 1 μM) or necroptotic VSMCs (TSZ-treated; TSZ: TNFα, SM-164 and Z-VAD-FMK, 1000X) respectively. **K** A schematic representation of study design to determine whether senescent VSMCs induces ferroptosis via releasing SASP factors. **L** Relative intracellular levels of MDA and GSH/GSSG ratio in VSMCs cultured with the transferred medium. $n = 8$ (for MDA) or 6 (for GSH/GSSG ratio) biologically independent samples. **M** Representative images of C11-BODIPY probe for evaluation of lipid peroxidation in VSMCs cultured with the transferred medium. Scale bar, 100 μm. Data expressed the mean ± SEM. The experiments in (**F**), (**H–J**), and (**M**) were repeated three times, and the representative images were presented. Comparisons of parameters were performed with two-sided unpaired $t$-test (**A–D**) or One-Way ANOVA followed by a Tukey's multiple comparisons test (**L**). Source data are provided as a Source Data file.

glutathione depletion induces ferroptosis and senescence[55]. Nevertheless, the exact causality relationship between pro-ferroptosis signaling and senescence remains unclear. Intriguingly, we found that SASP factors released from senescent VSMCs seemed to be unable to induce pro-ferroptosis signaling, whereas the secretary factors released by cells undergoing ferroptosis are sufficient to trigger cell senescence. Thus, our findings support the notion that ferroptosis signaling is a driver of vascular senescence and aging, but not vice versa. The exosomes or extracellular vesicles (EVs), may participate in this process. Ferroptotic macrophages released EVs to induce tumorigenesis[56]. We recently demonstrated that a skeletal muscle-derived EVs could be released into blood in response to physical exercise, and exerted anti-aging action on vasculature[24]. Thus, the cell-cell interactions between ferroptotic cells and adjacent cells is an interesting question needs future investigation.

Our findings support a notion that the activation of sub-lethal signals, such as the pro-ferroptosis signaling presented in this study, as well as the recently reported pro-apoptotic stress[57], and pro-pyroptosis stress[58], may not inevitably lead to cell death but could induce senescence. While senescence is canonically characterized as a state of permanent cell cycle arrest, recent advancements challenge the traditional understanding of senescence. For instance, p16[INK4A+] senescent cells in the basement membrane form a reparative niche in the lung[59]. Senescent cells hyperactivate hair growth by enhancing the activity of adjacent intact stem cells[60], and confer protection against type 1 diabetes in pancreatic β cells[61]. Senescent cells promote newt limb regeneration by promoting muscle dedifferentiation[62]. Our recent findings also demonstrate that the activated ferroptotic signaling in injured arteries facilitates VSMC dedifferentiation[63]. These findings suggest that senescence might not be an irreversible 'quiescent' state and thus there seems to be some interconversions between senescence and other intracellular phenomena, including cell death. The potential intricate regulatory patterns between cell death and senescence raise intriguing questions that require further investigation.

In addition to the above key findings, our results also add evidence in the understanding of ferritinophagy. Recent evidence has depicted the possible crosstalk between ferroptosis and autophagy, another type of conserved regulated cell death to maintains cellular homeostasis via the degradation of intracellular contents. Ferritinophagy, a selective form of autophagy, largely contributes to ferroptosis process. In ferritinophagy, NCOA4 delivers ferritin to lysosomes as a cargo receptor, and ferritin is then degraded to release $Fe^{2+}$[40], which subsequently causes excessive lipid peroxidation to induce membrane damage and cell death[64]. As we observed that ferritinophagy was blocked by GPX4 knockin or ferroptosis inhibitor, pro-ferroptosis signaling might also prime ferritinophagy. Therefore, a bidirectional association between ferroptosis and ferritinophagy may

exist, which further amplifies the generation of lipid peroxidation. From this viewpoint, disrupting this bidirectional loop between ferritinophagy and ferroptosis may serve as an ideal strategy to inhibit ferroptosis-related cell injury. NCOA4 forms insoluble condensates via multivalent interactions generated by the binding of iron to its intrinsically disordered region[42]. NCOA4 also acts not only as a canonical autophagy receptor but also as a driver to form ferritin liquid-liquid phase separation to facilitate the degradation of these condensates[43]. As disruption of the self-interaction of NCOA4 impairs ferritin degradation[43], whether the PPARγ-NCOA4 interaction affects NCOA4 self-oligomerization is still an open question. Considering the two studies have revealed important and complex regulation on NCOA4 properties[42,43], the regulatory mechanisms of NCOA4 on ferritin turnover may need more detailed investigation.

In our study, activation of cytoplasmic PPARγ blocked the ferritinophagy process and senescence. It is well-established that the co-factors PGC-1α/β are crucial for the nuclear function of PPARs[45]; however, we found that PGC-1α/β were not involved in the regulation of ferroptosis signaling on ferritinophagy, suggesting that PPARγ regulates NCOA4-centered ferritinophagy in a transcription factor-independent manner. As we observed that PPARγ was downregulated in nucleus but upregulated in cytosol, it is reasonable to propose the cytoplasmic PPARγ contributes to ferritinophagy regulation via interacting NCOA4. It should be noted that although NCOA4 always functions as a nuclear receptor, it was reported to regulate ferritinophagy in cytoplasm[40]. Our immunofluorescence colocalization study showed that the interaction of NCOA4 and PPARγ occurred at cytoplasm. PLA results also apparently support this notion, as it showed that the interaction between NCOA4 and FTH1 occurred outside the nuclei. An early study published in 1999 has reported an interaction between PPARγ and NCOA4 (called as ARA70 at that time)[48]; however, the biological function of this interaction is unknown yet. Our findings indicate that PPARγ competes with ferritin in binding to NCOA4: NCOA4 binds to ferritin via its C-terminal and binds to PPARγ via N-terminal. This complex is dynamic in response to extracellular stimuli and intracellular iron demand. These findings may advance the current understandings of PPARγ. Recently, Venkatesh et al. reported that PPARα is required for negative regulation on MDM2 and MDMX during ferroptosis execution[65]. These observations, along with our results, imply that PPAR family proteins may play key roles in ferroptosis regulation.

Our results may have potential translational implications for pharmacological therapeutics for CVD. Chronic senescence and senescence-associated SASP in vascular wall result in vascular calcification, arterial stiffening, and CVD[66]. Our findings demonstrate that either ferroptosis inhibitor or PPARγ agonist, have potential to be new senescence antagonists. In our work, ferroptosis inhibitor liproxstatin-1 displayed a potent antagonism against vascular

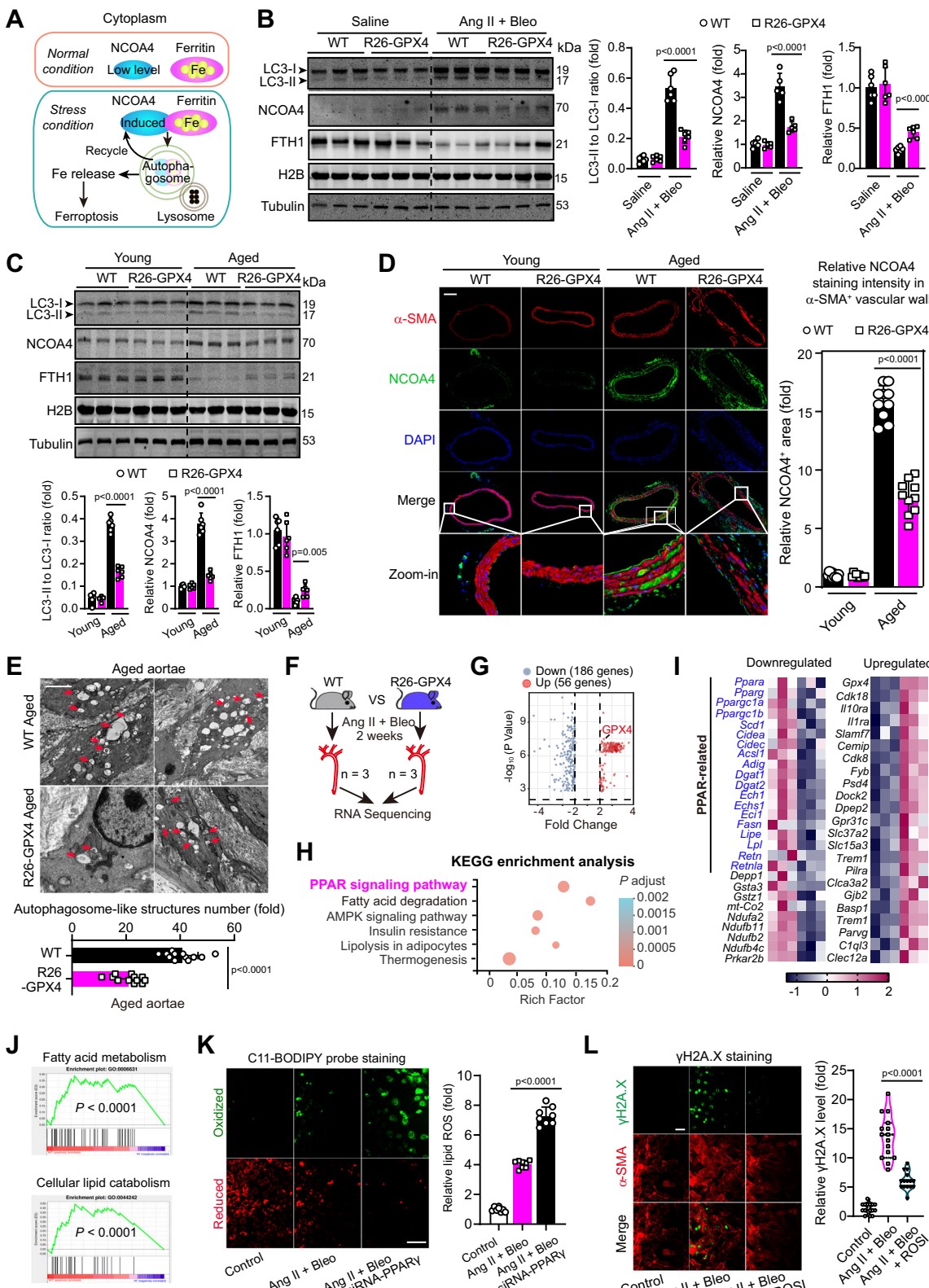

remodeling and stiffness. Interestingly, inhibition of ferroptosis signaling suppressed the induction of NAD$^+$ consumers (CD38 and PARP-1) in aging/senescent conditions. While NAD$^+$ repletion or removing NAD$^+$ consumers have been shown to inhibit senescence and enhance life/health span in multiple model organisms[9,10], the involvement of pro-ferroptotic signaling on NAD$^+$ consumers homeostasis is an intriguing topic needs future investigation. In

terms of PPARγ agonist, we believe our findings support the anti-senescence action of PPARγ agonist. In fact, PPARγ is an 'old' drug target and PPARγ agonist thiazolidinediones are used to treat type 2 diabetes mellitus for many years. Our findings not only for the first time point out the role of PPARγ in ferritinophagy, but also provide additional support for the anti-senescence action of PPARγ agonist.

**Fig. 6 | Inhibition of ferroptosis signaling suppresses ferritinophagy via regulating PPARγ signaling pathway. A** Current understanding of ferritinophagy process. **B** Immunoblotting analyses of autophagy marker LC3 and ferritinophagy-specific markers (NCOA4 and FTH1) in aortae. NCOA4 is a ferritinophagy cargo receptor and FTH1 is a substrate of ferritinophagy. Histone 2B (H2B) was used to demonstrate that full lysis has been achieved. $n = 6$ biologically independent samples. **C** Immunoblotting analysis of LC3, NCOA4, and FTH1 in aortae from young (2-month-old) or aged (24-month-old) WT and R26-GPX4 mice. $n = 6$ biologically independent samples. **D** Fluorescent immunohistochemistry showing the NCOA4 accumulation in aortae. α-SMA was used to indicate smooth muscle cells. DAPI was used to stain nuclei. Scale bar, 200 mm. $n = 10$ biologically independent samples. Scale bar, 100 μm. **E** Electron microscopy showing the number of autophagosome-like vacuoles. Scale bar, 1 μm. $n = 12$ biologically independent samples. **F** A schematic representation of study design showing the RNA-sequencing analysis in endothelial-denuded aortae from mice. **G** Volcano Plot of RNA-sequencing data showing the 186 upregulated genes and downregulated 56 genes (criteria: fold change absolute value > 2 and $P$ value < 0.01) in aortae from R26-GPX4 mice compared with WT mice under Ang II+Bleo condition. **H** KEGG analysis showing the differentially expressed genes. **I** Heatmap showing the top upregulated and downregulated genes in GPX4 knockin mice compared with WT mice. **J** GSEA analyses revealed that GPX4 knockin significant downregulated gene sets. **K** C11-BODIPY probe was used to assess the lipid ROS content in normal or PPARγ-knockdown VSMCs upon Ang II+Bleo-induced senescence stress. $n = 8$ biologically independent samples. Scale bar, 50 μm. **L** The influence of PPARγ activation on senescence was determined by immunofluorescent analysis with γH2A.X (green) in VSMCs under Ang II+Bleo-induced senescence stress. Scale bar, 100 mm. $n = 15$ biologically independent samples. Scale bar, 50 μm. Data expressed as mean ± SEM. Comparisons of parameters were performed with One-Way ANOVA followed by two-sided unpaired $t$-test in (**B**–**E** and **G**, **H**) or Tukey's multiple comparisons test in (**K** and **L**). Source data are provided as a Source Data file.

There might be several limitations in our investigation. Firstly, it should be noted that the vascular aging process is influenced not only by vascular smooth muscle cells (VSMCs) but also by other vascular cells, particularly endothelial cells. As Ang II has been reported to induce ferroptosis in endothelial cell[67], it is highly likely that ferroptosis in the endothelium plays a role in the aging process of the vasculature. It is an interesting question whether the interaction between ferroptosis in endothelial cells and VSMCs causes a synergistic effect in driving vascular stiffness/aging. Secondly, we utilized SM22α-driven AAV to specifically deliver GPX4 to VSMCs. However, viral infection may also occur in arteries of other organs such as lung and kidney, which could potentially alter the systemic vasculature status and further affect aortic aging. Lastly, our study employed SA-β-gal staining, as well as the assessment of expression levels of γH2A.X, p16, and p21, which are widely recognized markers for senescent cells at the bulk transcript level. Recently, advanced techniques at the single-cell level, such as scRNA-seq, have emerged as powerful tools for investigating the heterogeneity of cellular senescence[68,69] The absence of a single-cell level analysis in assessing the heterogeneity of cellular senescence might be a limitation of our study. Further investigation into the relationship between ferroptosis signaling and senescence, based on a single-cell level study, is warranted.

In conclusion, we demonstrate that pro-ferroptosis signaling drives vascular NAD⁺ loss, senescence, remodeling, and stiffness by promoting NCOA4-centered ferritinophagy, while inhibition of ferroptosis signaling or activation of PPARγ is sufficient to delay vascular senescence and aging. These findings indicate that targeting pro-ferroptosis signaling may be a promising strategy for treatment of senescence- or aging-associated CVD.

## Methods
### Animals
Eight-week-old C57BL/6J mice were obtained from Sino-British SIPPR/BK Lab Animal Ltd. (Shanghai, China). The knockin mouse stains harboring GPX4 at ROSA26 locus (R26-GPX4, project No. KICMS181221LY4) was generated by Cas9/CRISPR-mediated genome editing (Cyagen Biosciences, Santa Clara, CA, USA). For R26-GPX4 mice generation, the "CAG-mouse GPX4 cDNA (NM_008162.3)-polyA" cassette was inserted into intron 1 of ROSA26 under the guide of gDNA (gDNA, GGCAGGCTTAAAGGCTAACCTGG). The gRNA to mouse ROSA26 gene, the DNA containing mouse GPX4 gene, and Cas9 mRNA was co-injected into fertilized mouse eggs to generate targeted knockin offspring. F0 founder animals were identified by PCR followed by sequence analysis (PCR Primers: Forward, 5'-AAAGATCGCTCTCCACGCCCTAG-3'; Reverse, 5'-AGATGTACTGCCAAGTAGGAAAGTC-3'). The successful knockin of GPX4 allele was confirmed by sequencing with the primers as follows: Primer 1, 5'-CACTTGCTCTCCCAAAGTCGCTC-3'; Primer 2, 5'-ATACTCCGAGGCGGATCACAA-3'. The positive targeting was also confirmed southern blotting. The positive mice were bred to WT mice to test germline transmission and F1 animal generation was successfully established. For genotyping, the following primers were used: Forward, 5'-ACCTTTCTGGGAGTTCTCTGCTG-3'; Reverse 1, 5'-TACTTGGCATATGATACACTTGA-3'; Reverse 3, 5'-TTGTGGTGTATGTAACTAATCTG-3'. These mice were backcrossed for at least six generations with C57BL/6J mice. During the animal experiments, the genotypes of all investigated mice were known to investigator. No pre-established selection criteria for mice were used, other than gender and age. Animals were assigned randomly through a table of random numbers to cohorts by a technician that was blinded to the appearance or other characteristics of the animals. All operations in mice were approved by the Animal Care and Use Committee of Naval Medical University and followed the Principles of Laboratory Animal Care published by the National Institutes of Health (NIH publication 86-23 revised 1985) and ARRIVE guidelines. The mice were bred and housed under specific pathogen free conditions in the central animal facility. All mice were housed at a temperature of 21 °C–23 °C with relative humidity of 35%–65% and 12 h light/dark cycle in individually ventilated cages with access to water and standard chow diet.

### Human carotid artery samples
For evaluation the influence of ferroptosis in aged vascular wall, 'normal' carotid artery tissues from 6 middle-aged patients (<45 years old) and 6 elderly patients (>65 years old) underwent carotid artery aneurysm resection were obtained from Department of Vascular and Endovascular Surgery, Changzheng Hospital Affiliated to Naval Medical University between 2018 and 2021. The carotid aneurysm lesion was resected and a small section of adjacent relative 'normal' artery was resected simultaneously for further analysis. Internal carotid artery reconstruction was performed subsequently. Regarding the use of human samples, this study was performed according to the requirements of the Ethical Committee of Changzheng Hospital and the Declaration of Helsinki. All participants gave written informed consent.

### Primary human carotid artery VSMCs culture
Primary human carotid artery VSMCs were obtained in arteries tissue from patients underwent carotid aneurysm resection and repair surgery without atherosclerosis. Regarding the use of human samples, this study was performed according to the requirements of the Ethical Committee of Changzheng Hospital and the Declaration of Helsinki. All participants gave written informed consent. After removing the endothelial layer and adventitia from the aortic tissue swiftly, the carotid artery tissues were cut into 2 mm² pieces, placed into 6-well plates containing 1 ml cultured media and grown for ~2 weeks to allow the cells to emerge. The primary human VSMCs were cultured in a low serum (5% fetal bovine serum) human VSMC-specific medium (Promocell, Smooth Muscle Cell Growth Medium 2, C22062) supplemented with 1% penicillin-streptomycin (15140122, Gibco, MA, USA) at 37 °C in a humidified incubator with 5% $CO_2$. These human VSMCs were

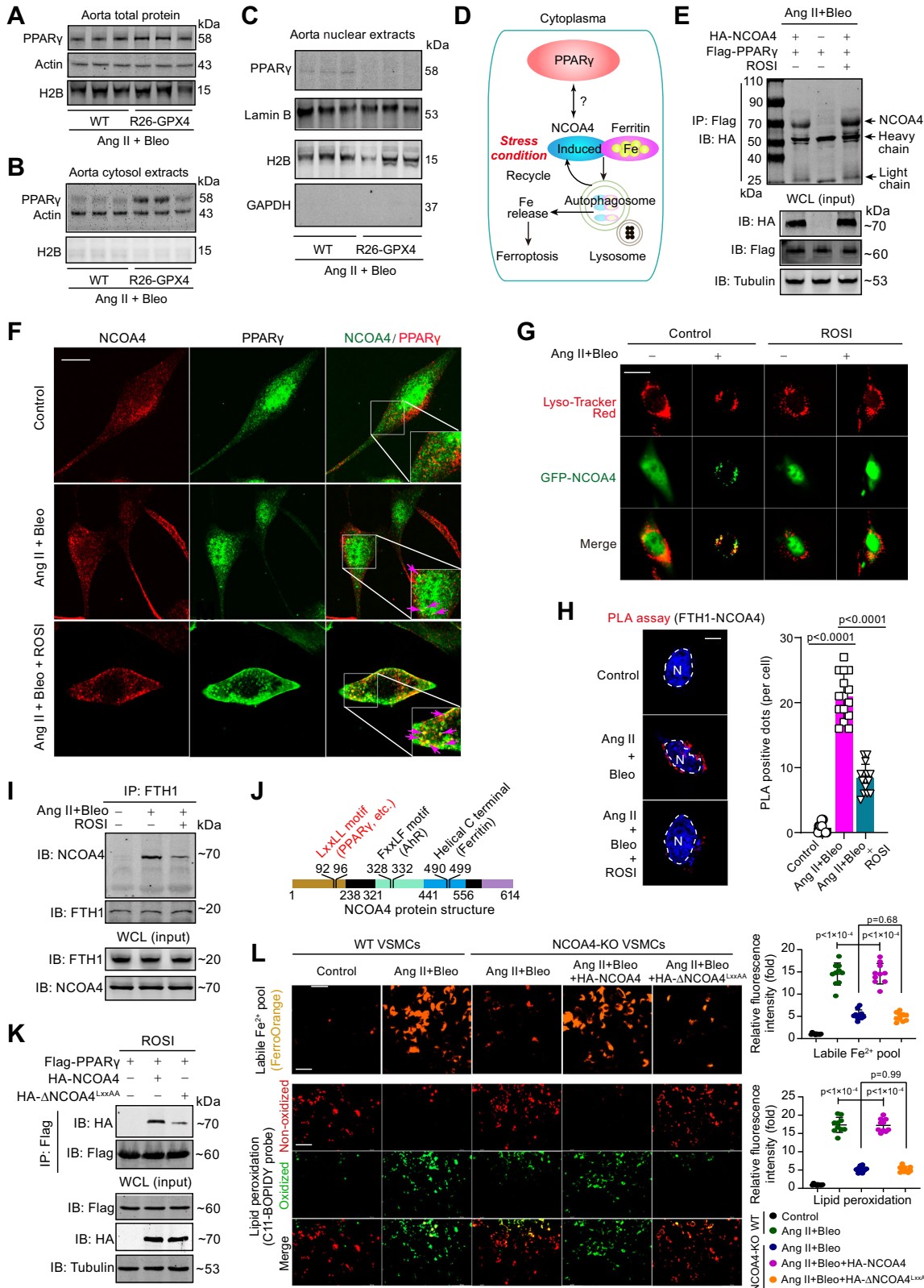

used to evaluate the pro-ferroptosis signaling in replicative senescent condition.

## Measurement of circulating ferroptosis-related molecules and arterial stiffness in human

The cohort study in this work aimed to include 80 voluntarily attended participants who underwent regular health examination in Shanghai Tenth People's Hospital affiliated Tongji University Medical School from Dec 2018 to Dec 2019 as described previously[24]. Finally, 73 people were included in present analysis with the response rate 91.3%. The inclusion criteria were: (1) over 18 years old; (2) stable on drugs and lifestyle, which means no significant changes in medication, diet, or exercise habit for at least one month; (3) willing and able to complete the study with written informed consent. The exclusion

**Fig. 7 | Cytoplasmic PPARγ interacts with NCOA4 to disassociate NCOA4-ferrtin ferritinophagic complex. A–C** Immunoblotting analysis of PPARγ protein level in total protein (**A**), cytoplasmic fraction (**B**), and nuclear fraction (**C**) of aortae from WT mice and R26-GPX4 mice under Ang II+Bleo-induced senescence stress. **D** A proposed action model of cytoplasmic PPARγ in ferritinophagy. **E** Interaction between Flag-PPARγ and HA-NCOA4 was detected by co-immunoprecipitation and immunoblotting in cultured VSMCs. Rosiglitazone (ROSI, 10 μM) was used to activate PPARγ. **F** Immunofluorescence colocalization analysis of NCOA4 and PPARγ in cultured VSMCs upon Ang II+Bleo stress or rosiglitazone (ROSI, 10 μM). Scale bar, 10 μm. **G** Immunofluorescence analysis showing that the ferritinophagy process triggered by Ang+Bleo-induced senescence stress in cultured VSMCs was blocked by PPARγ activator rosiglitazone (ROSI, 10 μM). VSMCs were transfected with GFP-tagged NCOA4 and ferritinophagy process was monitored by immuno-fluorescent analysis with Lyso-Tracker and GFP-tagged NCOA4. Scale bar, 20 μm. **H** In situ proximity ligation assay (PLA) showing the interaction between FTH1 and NCOA4 in cultured VSMCs. DAPI was used to stain nuclei. $n = 15$ biologically independent samples. Scale bar, 10 μm. **I** The influence of PPARγ activation on interaction between NCOA4 and FTH1 was evaluated by co-immunoprecipitation and immunoblotting. **J** Schematic representation of NCOA4 protein structure. The LxxLL motif responsible for the interaction between NCOA4 and PPARγ is highlighted in red. **K** Interaction between PPARγ and wild-type NCOA4 or mutant NCOA4 (ΔNCOA4$^{LxxAA}$) was evaluated using co-immunoprecipitation and immunoblotting in VSMCs. **L** Assessment of liable Fe$^{2+}$ pool by FerroOrange probe and lipid peroxidation by C11-BOPIDY probe under fluorescence microscope in normal and NCOA4-null VSMCs upon Ang II+Bleo stress. In NCOA4-null VSMCs, the wild-type NCOA4 or mutant NCOA4 (ΔNCOA4$^{LxxAA}$) were overexpressed for investigation the importance of LxxLL motif in NCOA4 on ferroptosis. $n = 10$ biologically independent samples. Scale bar, 20 μm. Data expressed as mean ± SEM. The experiments in (**A–C**), (**E–G**), and (**I–K**) were repeated three times, and the representative images were presented. Comparisons of parameters were performed with One-Way ANOVA followed by a Tukey's multiple comparisons test. NS no significance. Source data are provided as a Source Data file.

criteria were: (1) severe organ dysfunctions including heart (NYHA III-IV), kidney (CKD 4-5), or liver (Child-Pugh score >6); (2) musculoskeletal diseases; (3) physical disability relevant to routine exercise; (4) severe uncontrolled hypertension (resting systolic/diastolic blood pressure >180/110 mmHg) or diabetes (HbA1c > 10.0%); (5) recent cerebral or cardiovascular events including myocardial infarction and stroke; (6) low life expectancy like malignant tumor.

Basic information includes age, gender, smoking, drinking, etc., was collected with standard structural questionnaire or previous medical records. Serological indicators like fasting blood glucose, creatinine, cholesterol, etc., were tested in clinical laboratory of Shanghai Tenth People's Hospital affiliated Tongji University Medical School. Exercise status was assessed based on self-reported results to several fixed questions including exercise frequency (times per week, zero indicating no exercise) and exercise time (time for each exercise on the average). The carotid-femoral pulse wave velocity (cfPWV) was measured with SphygmoCor device (AtCor Medical, Australia) by two experienced operators as described previously[24]. Blood was drawn into pyrogen-free tubes with EDTA and the plasma samples were stored at −80 °C for measurement of 15-HETE (#534721, Cayman Chemical, USA) and MDA (Abcam) using commercial kits. All examinations were performed with the informed consent of the subjects. The protocol of the study was conducted in accordance with the Declaration of Helsinki and was approved by the Ethics Committee of Shanghai Tenth People's Hospital affiliated Tongji University Medical School. All participants gave written informed consent.

### Measure of pulse wave velocity (PWV) in mice
A high-resolution Vevo 2100 system (VisualSonics Inc., Toronto, Canada) was used for ultrasound imaging of aorta and measure PWV as described previously[24]. In brief, the mice were first anesthetized with 3.0% isoflurane mixed with oxygen (100%, airflow velocity: 1 L/min) to maintain a surgical plane of anesthesia. The heart rate was kept to 400–500 beats per min. A high-frequency array probe with a center frequency of 40 MHz (Vevo MS550D) was used to collect B-mode images. Pulse wave at the aortic arch and abdominal aorta was recorded. The PWV was calculated by dividing the arterial distance between the two points and the time delay for the pulse which was calculated based on the ECG.

### Mouse blood pressure measurement
Both noninvasive tail blood pressure was measured as described previously[24]. Animals were anesthetized (pentobarbital, 40 mg/kg, i.p) during measurement and tuff-based tail monitor (BP-2000, Visitech Systems, Apex, NC) was used to record tail blood pressure. Pulse pressure (PP) was calculated as the difference between systolic and diastolic blood pressure.

### Mouse experimental vascular senescence model
The vascular stress mouse model was induced by infusion of angiotensin II (Ang II, 400 ng/kg/min), a well-characterized hormone related to vascular malfunction, remodeling and senescence[21], plus a pro-senescent DNA-attacking agent bleomycin (Bleo, 40 ng/kg/min)[25,26] for 2 weeks using Alzet® micro-osmotic pump (#1004, Alzet) as descried in our previous studies[23,24]. For blockade of ferroptosis signaling, liproxstain-1 was injected intraperitoneally (5 mg/kg/d).

### Mouse natural vascular aging model
The aortae of naturally aged mice were considered to be the vasculature with natural vascular aging. These aortae were isolated from naturally old mice (24-month-old). The aortae isolated from young mice (2-month-old) were used as control.

### Tissue and blood sampling
The mice were anesthetized with pentobarbital sodium (40 mg/kg, intraperitoneally) and the blood was drawn. Aortae were dissected carefully and washed in ice-cold PBS solution for two times, and then fixed in 10% formalin for further examination. Serum was obtained by centrifugation of the blood at $1000 \times g$ for 5 min.

### Mouse VSMCs culture
Mouse vascular smooth muscle cell line (MOVAS) obtained from American Type Culture Collection (ATCC) were used for other biochemical experiments. The MOVAS cells were cultured in DMEM medium (Gibco, MA, USA) containing 10% fetal bovine serum (FBS, 10099141, Gibco, MA, USA) and 1% penicillin-streptomycin (15140122, Gibco, MA, USA).

### Cell treatment
For inhibition of ferroptosis, liproxstatin-1 (final concentration 1 μM, S7699, Selleck) was added into the culture medium. To monitor the ferroptosis activator on senescence, RSL3 at a non-cytotoxic dose (final concentration 0.5 μM, Selleck) and erastin was added into culture medium for 5 days. To induce ferroptosis, RSL3 at a cytotoxic dose (10 mM, Selleck) and another ferroptosis activator erastin at a cytotoxic dose (20 mM, Selleck) were used in cultured VSMCs. To induce apoptosis, VSMCs were treated by ABT-737 (1 μM, Beyotime). To induce necroptosis, VSMCs were treated by TSZ complex (1000X, Beyotime). PPARγ antagonist T0070907 and activator rosiglitazone were purchased from Selleck.

### Cell senescence model
For studying replicative senescence in vitro, human VSMCs were cultured until replicative senescence was reached (passage > 35). The human VSMCs cultured at passages 2–10 was thought to be

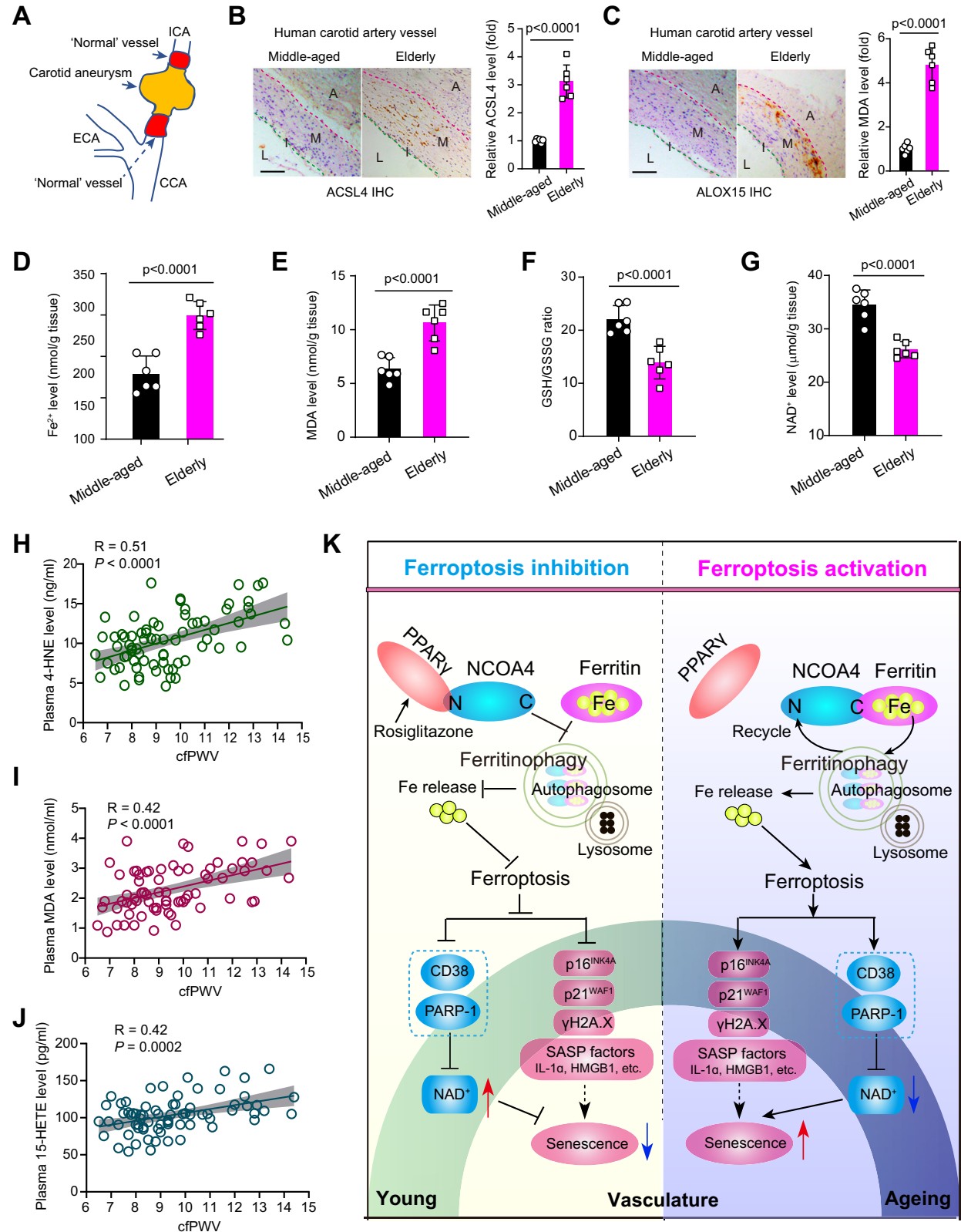

normal control. For studying stress-induced senescence in vitro, MOVAS cells were starved at 0.1% FBS DMEM medium overnight to achieve synchronization and then stimulated by recombinant Ang II (0.1 μM, #A6778, Sigma) plus bleomycin (100 nM, S1214, Selleck) for 5 days. The culture medium was changed every day to prevent Ang II degradation.

**Senescence-associated-β-galactosidase (SA-β-gal) activity staining**

SA-β-gal activity staining was performed with a commercially available kit (#C0602, Beyotime Biotechnology, China). Cultured cells or aortae samples were washed in ice-cold PBS buffer for three times and then fixed in 1% glutaraldehyde for 15 min at room temperature. After

**Fig. 8 | Activation of pro-ferroptosis signaling is associated with arterial aging/ stiffness in human.** **A** A surgical illustration for the harvest of 'normal' arterial tissue during carotid aneurysm resection. The carotid aneurysm was resected and a small section of adjacent relative normal artery was resected simultaneously. Internal carotid artery reconstruction was performed subsequently. ICA internal carotid artery, ECA external carotid artery, CCA common carotid artery. **B**, **C** Immunohistochemistry analysis showing the two ferroptosis signaling markers ACSL4 (**B**) and ALOX15 (**C**) in carotid arterial tissues from middle-aged (<45-y-old) and elderly (>65-y-old) patients. $n = 6$ biologically independent samples. Scale bar, 100 μm. **D**, **E** Levels of ferroptosis initiator $Fe^{2+}$ (**D**) and lipid peroxidation maker MDA (**E**) in carotid arterial tissues from middle-aged and elderly patients. $n = 6$ biologically independent samples. **F**, **G** GSH/GSSG ratio (**F**) and NAD⁺ level (**G**) in carotid arterial tissues from middle-aged and elderly patients. $n = 6$ biologically independent samples. **H**–**J** Associations of arterial stiffness index (carotid-femoral PWV [cfPWV]) and plasma levels of ferroptosis-related molecules, including 4-HNE (**H**), MDA (**I**) and 15-hydroxyeicosatetraenoic acid (15-HETE, **J**). The cfPWV was measured using SphygmoCor® technology in human volunteers. The concentrations of 4-HNE, MDA, and 15-HETE were determined using commercial kits. $n = 73$ biologically independent samples. **K** A proposed schematic model of the regulatory effects of pro-ferroptosis signaling on senescence in VSMCs. Under stress, ferritinophagy is activated to release $Fe^{2+}$ and induce pro-ferroptosis signaling, which lead to senescence by upregulating of pro-senescent molecules such as p16[INK4A]/ p21[WAF1], γH2A.X, and SASP factors, as well as inducing of NAD⁺ consumers CD38/ PARP-1. VSMCs senescence promotes vascular stiffness and accelerates vascular aging. Data expressed as mean ± SEM. Comparisons of parameters were performed with two-sided unpaired $t$-test (**B**–**G**) or linear correlation analysis (**H**–**J**). Source data are provided as a Source Data file.

washing for three times, the cells or aortae were incubated in staining buffer (X-gal concentration: 1 mg/ml) for 12-h (for cells) or 24 h (for aortae) at 37 °C. The images were obtained with light microscope (cultured cells) or camera (tissue samples), and then analyzed with ImageJ software (NIH, Bethesda). The degree of the senescence (blue-green) was evaluated according to the grading score system.

### Immunoblotting
Tissue samples were lysed with RIPA lysis buffer (#P0013B, Beyotime) containing 50 mM Tris (pH 7.4), 150 mM NaCl, 1% Triton-X-100, 1% sodium deoxycholate, 0.1% SDS and protease/phosphatase inhibitor cocktails (#1046, Beyotime) in a homogenizer at 4 °C. Cultured cells were directly lysed on ice with the buffers. Proteins were run on 8-12% SDS-PAGE gels at 120 V for 1.5 h at room temperature. Then gels were transferred onto nitrocellulose membranes at 200 mA for 1 h in ice-water bath. Membranes were blocked with 5% non-fat milk dissolved in PBST buffer (0.5% v/v Tween-20 in PBS buffer) for 2 h at room temperature. Membranes were washed (5 min × 3 times) with PBST, and incubated with primary antibodies (anti-GPX4, #ab125066, Abcam; anti-ACSL4, #sc-365230, Santa Cruz Biotechnology; anti-ALOX15, #PA5-15065, Invitrogen; anti-TFR1, #10084-2-AP, Proteintech; anti-MDA, #ab6463, Abcam; anti-4-HNE, #ab46545,Abcam; anti-p53, #A5761, Abclonal; anti-p16[INK4A], #sc-1661, Santa Cruz Biotechnology; anti-p21[WAF1], #sc-817, Santa Cruz Biotechnology; anti-γH2A.X[Ser139], #ab81299, Abcam; anti-H2A.X, #A11412, ABclonal; anti-PARP-1, #13371-1-AP, Proteintech; anti-CD38, #sc-374650, Santa Cruz Biotechnology; anti-IL-1α, # 16765-1-AP, Proteintech; anti-HMGB1, #ab18256, Abcam; anti-LC3, #14600-1-AP, Proteintech; anti-NCOA4, #sc-373739, Santa Cruz Biotechnology; anti-FTH1, #4393, Cell Signaling Technology; anti-PPARγ, #16643-1-AP, #60004-1-Ig, Proteintech; anti-tubulin, #AC008, ABclonal) overnight at 4 °C. The dilution of primary antibodies ranged from 1:1000 to 1:5000. After washing, membranes were incubated with secondary antibodies conjugated with IRDye ®800CW (Li-Cor Biosciences). Images were photographed with Odyssey system (Li-Cor Biosciences) and then analyzed with ImageJ software (NIH).

### Immunofluorescence and immunohistochemistry
In immunofluorescence assay, cells or tissues were fixed in cold 4% formaldehyde for at least 30 min and treated with 0.1% Triton-X-100, and then incubated with primary antibodies (anti-GPX4, #ab125066, Abcam; anti-ACSL4, #sc-365230, Santa Cruz Biotechnology; anti-ALOX15, #PA5-15065, Invitrogen; anti-MDA, #ab6463, Abcam; anti-4-HNE, #ab46545,Abcam; anti-p53, #A5761, Abclonal; anti-p16[INK4A], #sc-1661, Santa Cruz Biotechnology; anti-p21[WAF1], #sc-817, Santa Cruz Biotechnology; anti-γH2A.X[Ser139], #ab81299, Abcam; anti-IL-1α, # 16765-1-AP, Proteintech; anti-HMGB1, #ab18256, Abcam; anti-NCOA4, #sc-373739, Santa Cruz Biotechnology; anti-FTH1, #4393, Cell Signaling Technology; anti-α-SMA, #14395-1-AP, Proteintech) overnight at 4 °C. The dilution of primary antibodies ranged from 1:200 to 1:1000. On the next morning, the sections were incubated with secondary antibodies conjugated with fluorophores Alexa Fluor 488 or Alexa Fluor 555 dyes (Invitrogen) for 1 h at room temperature. Nuclei were counterstained in DAPI (Invitrogen). The images were captured by confocal microscope FluoView™ FV1000 or digital microscope (Leica Microsystems, Berlin, Germany). In immunohistochemistry, mouse aorta was fixed in 4% formaldehyde plus 4% sucrose for 20 min and then dehydrated in 30% sucrose. After embedded in paraffin, sections (8 μm) were blocked in 5% normal goat plasma for 3 h at room temperature. The sections were incubated with primary antibodies overnight at 4 °C and followed by peroxidase-conjugated secondary antibodies (Proteintech). 3,3-diaminobenzidine was used as a chromogenic reagent. Nuclei was stained with hematoxylin. Coverslips were imaged with a digital microscope (Leica Microsystems, Berlin, Germany). To detect ferroptosis and senescence, we used a combination of ferroptosis probe (Liperfluo probe[15]) and senescence' probe KSAP1[38] to stain VSMCs. Images were obtained with a confocal microscope (FV1000, Olympus). All the image analyses were performed with ImageJ software (NIH).

### Histology staining
Elastica-van-Gieson (EVG), Masson's trichrome, Alizarin Red staining were used to assess elastin degradation, collagen-related fibrosis, and calcium deposit respectively. The aorta sections (8 μm) were stained with standard procedures.

### Elastin Staining and Degradation
Briefly, suprarenal aortic samples from the different groups of mice were embedded in paraffin, cut, and then measured with EVG staining. We used ×40 magnification to evaluate elastin degradation. An established standard for the elastin degradation score was adopted as follows: score 1, no elastin degradation, well-organized elastin lamina; score 2, mild elastin degradation with some interruptions or breaks in the lamina; score 3, moderate elastin degradation with multiple interruptions or breaks in the lamina; and score 4, severe elastin fragmentation or loss or aortic rupture.

### Immunoprecipitation
The normal MOVAS cells or MOVAS cells with PPARγ-knockdown were treated with Ang II+Bleo for 5 days and were lysed in NP-40 buffer with protease inhibitor cocktail and the crude lysates were cleared of insoluble debris by centrifugation at $12,000 \times g$. The lysates were immunoprecipitated with normal IgG or primary antibodies, including anti-FTH1 (#4393, Cell Signaling Technology), anti-HA-tag (#ab9110, Abcam) and anti-Flag-tag (#ab205606, Abcam) in a rotator (Scilogex, Rocky Hill, CT) at 4 °C overnight. The 20 μl protein A/G PLUS-Agarose beads (#sc-2003, Santa Cruz Biotechnology) were added into the 200 μl homogenates or lysates and incubated for 4 h with gentle agitation. The beads were washed 3 times with the lysis buffer and boiled with 10 μl loading buffer. The beads were removed by centrifugation (5 min at $12,000 \times g$). The supernatant fraction was collected and used for immunoblotting.

## Quantitative PCR

Total RNA of cells or tissues was extracted with TRIzol reagent (Invitrogen). The quality of RNA was assessed with 260/280 nm absorption ratio using Nanodrop (Thermo Fisher Scientific). RNA was used only when the ratio was 1.8-2.0. After RNA concentration measurement, 2 µg RNA was used in the reverse transcription with One-Step gDNA Removal and cDNA Synthesis SuperMix (TransGen Biotech, China). Real-time quantitative PCR (qPCR) was performed using Bio-Rad CFX96 system (Bio-Rad Laboratories, Hercules, CA) with SYBR Premix Ex Taq Mixture (Takara, Tokyo, Japan). Each sample was determined in duplicate and housekeeping gene GAPDH was used as the reference gene for normalization. Primer sequences were listed in Supplementary Table 2. Cycle threshold (CT) values were recorded and $^{\Delta\Delta}$CT method was used to quantify fold changes of genes.

## EdU staining assay

Cultured VSMCs were incubated with 10 µM EdU (included in the kit) for 2 h prior to harvesting. EdU-incorporated cells were detected by confocal microscopy after Click-iT® EdU labeling following the manufacturer's protocol (Click-iT™ EdU Alexa Fluor™ 594 Imaging Kit, Thermo Fisher Cat# C10339).

## Cell viability Assay

Cell viability assay was carried out using Cell Counting Kit-8 kit (CCK-8, Dojindo Laboratories, Kumamoto, Japan). VSMCs were plated in 96-well plates in triplicate at approximately $1 \times 10^5$ cells per well and cultured in the growth medium. The cell viability of cells per well were measured by the absorbance (450 nm) of CCK-8 solution at the indicated time points.

## Transcriptomics study

Transcriptomics study was conducted using bulk RNA-sequencing. The aortae from WT and R26-GPX4 mice were obtained after two weeks of infusion of Ang II+Bleo, and the total RNA was extracted from the aortae tissues without tunica intima, tunica adventitia, and perivascular adipose. The quality control of total RNA samples was performed with a 2100 Expert Bioanalyzer (Agilent) and RNA sequencing was performed with an Illumina Hiseq2000 platform of Majorbio Biotech (Shanghai, China). The data were analyzed online with I-Sanger Cloud Platform. The bulk RNA-sequencing raw data of aortae tissue from WT and R26-GPX4 mice treated with Ang II+Bleo have been deposited in the NCBI Sequence Read Archive (SRA) database (PRJNA907269).

## Plasmid constructs and transfection

The pcDNA3.1 plasmid was obtained from Thermo Fisher Scientific (#V79520). Plasmids carrying mouse GPX4, PPARγ, GFP-tagged NCOA4, HA-tagged NCOA4, HA-tagged ΔNCOA4$^{LxxAA}$ and Flag-PPARγ were generated based on the pcDNA3.1 plasmid backbone. The coding sequences of mouse GPX4, PPARγ and GFP-tagged NCOA4 were subcloned into pcDNA3.1 backbone. The pcDNA3.1 plasmid containing no inserted DNA sequence was used for as a control. Plasmids were extracted using an EndoFree Plasmid Maxi Kit (#12362, Qiagen, Germantown, MD). To ensure the correctness of the sequence, the plasmid was sequenced using a T7 promoter primer. The purified plasmids were transfected into MOVAS cells using Lipofectamine LTX Reagent with PLUS Reagent (Thermo Fisher Scientific, #15338100). MOVAS cells were grown to 50%–70% confluence in a 24-well culture dish and transfected with plasmids or control vector. For each well within a 24-well plate, 1 µg plasmid DNA was mixed with 2 µl LTX and 0.5 µl PLUS reagent and was then added into the culture medium. After overnight incubation (6-8 h), the cells were switched back to normal culture medium. Experiments were conducted at two days post-transfection. For knockout of NCOA4, a one plasmid system pX330-U6-Chimeric_BB-CBh-hSpCas9 (Addgene, Cambridge, MA) carried was

transfected into mouse MOVAS cells with Lipofectamine 3000 (L3000-015, Life Technology, Carlsbad, CA, USA) according to the manufacturer's protocol. Two days later, cells were treated with puromycin (2 µg/ml) for 5 days to select positive clones. The survived cells (NCOA4-null cells) were kept for culture and expanded. The targeting sequence were as follows: NCOA4-KO#1: CACGCGAGCTCCTCAAG-TATTGG; NCOA4-KO#2: CGTCGCTGATTGTTGCGCCGGGG.

## VSMC-specific delivery of GPX4 by adeno-associated virus (AAV)

The AAV carrying GPX4, mutant GPX4 with its catalytic site mutation from selenocysteine to cysteine (GPX4$^{Cys}$), and negative control AAV vector were constructed and packaged as described previously[63]. Briefly, the vector pHBAAV-SM22a-ZsGreen was selected for use in this study, and restriction endonuclease BamHI and HindIII was used for vector cleavage to obtain a purified linearized vector. The synthesized GPX4 or GPX4$^{Cys}$ fragments were amplified using 2xFlash PCR Master Mix (Dye) kit according to the manufacturer's instructions. An HB infusion-TM kit (Hanbio Biotechnology) was then used for the ligation of the linearized vector and mouse GPX4 fragments, according to the manufacturer's instructions, followed by transformation of DH5α competent cell (TIANGEN). After cultivating with LB medium in culture plates for 12–16 h, the bacterial solution was used for PCR identification with 2xHieff PCR Master Mix (Dye) kit (YEASON) according to the manufacturer's instructions and the amplified sequence was detected by Sanger Sequencing to verify the consistency with GPX4. The plasmid was extracted with TIANpure Mini Plasmid Kit (TIANGEN) according to the manufacturer's instructions. Finally, the extracted plasmid was co-transfected with Packaging plasmids (pAAV-RC and pHelper) into HEK-293T cells using LipofiterTM transfection reagent (HanBio) as per the manufacturer's instructions. After 72 h of transfection, the transfected HEK-293T cells were centrifuged and broken, and the supernatant was collected for virus purification using Vira-Trap™ AAV Purification Maxiprep Kit (Biomiga). The titer is $1.6 \times 10^{12}$ vg/mL. For gene delivery of GPX4 specifically in VSMCs, the mice were administrated by AAV for twice via tail vein injection with the indicated virus ($3 \times 10^{11}$ vg per mouse).

## Proximity ligation assay

The Proximity ligation assay was performed using Duolink™ In Situ Red Starter Kit Mouse/Rabbit (CAT#DUO92101-1KT, Sigma-Aldrich) following the manufacturer's protocol. Briefly, after permeabilization, cells were blocked against nonspecific binding with Duolink® Blocking Solution for 2 h at 37 °C. Anti-NCOA4 (CAT#sc-373739, Santa Cruz) and Anti-ferritin (CAT#4393, Cell Signaling Technology) were then incubated with cells in Duolink® Antibody Diluent overnight at 4 °C. After washed with wash buffer A, cells were incubated with Duolink® PLA Probe for 60 min at 37 °C. Cells were then incubated with the ligase and Duolink® Ligation buffer for 30 min at 37 °C. Next, cells were washed in wash buffer A for 10 min at room temperature (RT) and incubated with the polymerase and amplification buffer for 100 min at 37 °C. Finally, cells were washed in wash buffer B and 0.01× wash buffer B at RT. Then, cells were mounted with Duolink® In Situ Mounting Medium with DAPI and imaged with FV1000 confocal microscopy (Olympus).

## Iron level determination

The quantitation on the Fe$^{2+}$ levels in tissues or cells was based on a colorimetric assay kit from Sigma-Aldrich (#MAK025). The tissue or cells were lysed with the buffer and the Fe$^{2+}$ concentration was determined according to the manufacturer's instruction.

## Labile iron pool assay

The cells ($1 \times 10^5$) were seeded on 96-well plates and treated with Ang II+Bleo for 5 days. The cells were incubated with 1 µM FerroOrange probe (Dojindo, Japan) with 1 mg/mL BSA for 30 min at 37 °C. After washing by D-Hank's solution for three times, 100 µL of 1X D-Hank's

solution was added and the fluorescence intensity the excitation of 488 nm monitored under a fluorescence microscope digital microscope (Leica Microsystems, Berlin, Germany). The FerroOrange fluorescent intensity is proportional to the labile iron concentration.

## Evaluation of oxidative stress

Levels of MDA (#700870) and TAC (#709001) in tissue or cells were determined with commercial kits from Cayman Chemical (Arbor, MI). To specifically evaluate lipid ROS production in cells, the cells were incubated with C11-BODIPY dye (5 μM, Invitrogen, Molecular Probes) with for 30 min at 37 °C, and then washed with PBS by three times. The ROS intensity was measured using flow cytometry (FACS Calibur, BD Biosciences, Corp., San Jose, CA) or visualized by FV1000 confocal microscopy (Olympus).

## NAD$^+$ and GSH determination

The tissue or cells were lysed with extraction buffer and then the protein concentration in samples was determined using a BCA kit (Beyotime, China) according to the manufacturer's instructions. Then, the samples were deproteinized with spin column and the NAD$^+$ level was examined with a commercial NAD$^+$/NADH quantification kit (#ab65348, Abcam). The GSH level in tissue and cells was determined using a commercial kit utilizing an enzymatic recycling method, using glutathione reductase for the quantification of GSH (#703002, Cayman Chemical, Arbor, MI). The NAD$^+$ and GSH/GSSG ratio were calculated using the standard curve.

## Transmission electron microscopy

Aorta tissues were fixed with 2% paraformaldehyde and 2% glutaraldehyde in 0.1 mol/L phosphate buffer (pH 7.4), followed by postfixation for 8 h in 1.5% osmium tetraoxide. After dehydration with graded alcohols, the samples were dehydrated in a graded ethanol series and embedded in epoxy resin. Samples were sectioned (80 nm), counterstained with uranylacetate and lead citrate, and observed with a transmission electron microscope (Hitachi, H-800, Tokyo, Japan). Images were acquired digitally from a randomly selected area and the number of autophagosome-like structures was calculated.

## Ferritinophagy assay

Ferritinophagy was evaluated by several assays. The first is determination of NCOA4 and FTH1 protein levels with immunoblotting analysis. The aortic tissues isolated from mice or MOVAS cells were lysed and probed with anti-NCOA4 and FTH1. The second is the fusion of GFP-tagged NCOA4 and lysosomes. Briefly, MOVAS cells were transfected with GFP-tagged NCOA4 plasmid or siRNA-PPARγ. Five days later, these cells were incubated with Lyso-Tracker Red (50 nM, Invitrogen) for 1 h at 37 °C. The fluorescent images of Lyso-Tracker Red and GFP-NCOA4 was captured by FV1000 confocal microscope (Olympus, Tokyo, Japan). The NCOA4$^+$/lysosome$^+$ puncta-like structures are thought to be vesicles in ferritinophagy.

## Publica gene expression dataset analysis

Public gene expression data (GSE1011) was downloaded as raw signals from Gene Expression Omnibus (http://www.ncbi.nlm.nih.gov/geo), interpreted, normalized, and log2-scaled using the online analysis tool Bioinformatics websites (http://www.bioinformatics.com.cn and https://hiplot.com.cn/). Identification of differentially expressed gene sets between normal and aortae treated in GSE1011 was also performed using the Bioinformatics and Hiplot tools.

## Statistical Analysis

All results are shown as mean ± SEM. Data distribution within all separate groups were validated by the Shapiro–Wilk normality test. Unpaired two-tailed t-test was used for comparison of two groups, if the normality test was not passed the non-parametric Mann–Whitney test was used. One-Way ANOVA followed by Tukey's multiple comparison test was used for comparison of more than two groups, if normality test was not passed the non-parametric Kruskal–Wallis test was performed. The statistical significance level was set at 0.05, unless otherwise stated. All statistical analyses were performed with GraphPad Prism 8.

## Reporting summary

Further information on research design is available in the Nature Portfolio Reporting Summary linked to this article.

## Data availability

Source data for Fig. 1–10 and Supplementary Data Figs. 1–15 are provided as a Source Datafile. The raw RNA-sequencing data of aortae tissue from mice has been deposited in the NCBI Sequence Read Archive (SRA) database under accession PRJNA907269. The public gene expression dataset GSE1011 can be accessed through the Bioproject Database (https://www.ncbi.nlm.nih.gov/bioproject/PRJNA87077). Source data are provided with this paper.

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

## Acknowledgements
This work was supported by the grants from National Natural Science Foundation of China (91849135 [P.W.], 82073915 [P.W.], 81773719 [D.J.L.], 81973312 [F.M.S.] and 81971306 [D.J.L.]), National Key Research and Development Project (2018YFA0108301 [P.W.]), Shanghai Science and Technology Commission (21XD1424900 [P.W.] and 21S11901200 [F.M.S.]), Shanghai Shuguang Program (19SG32, [P.W.]) and Shanghai Municipal Three Year Action Plan for Strengthening the Construction of Public Health System (GWVI-11.2-XD06, [D.J.L.]). We thank the patients and their families. We thank Prof. Yuan Guo (College of Chemistry and Materials Science, Northwest University, China) for providing the fluorescent senescence probe KSA01.

## Author contributions
P.W., D.J.L., F.M.S., and L.F.Q. conceived and designed research; D.Y.S., S.X.O., C.C., Yu S., J.J.X., Q.X.J., J.H.M., J.T.F., J.T., W.B.W., P.P.Z., and J.B.Z. performed experiments; J.J.W and L.F.Q. provided material; D.Y.S., D.J.L., F.M.S., and P.W. analyzed data. P.W. and D.J.L. wrote the manuscript. J.J.W., Yi S., and Z.Y.L. discussed and edited the manuscript. P.W., D.J.L., and F.M.S. provided funding. All authors contributed with productive discussions and knowledge to the final version of this manuscript.

## Competing interests
The authors declare no competing interests.

## Additional information

[1]The Center for Basic Research and Innovation of Medicine and Pharmacy (MOE), School of Pharmacy, Second Military Medical University/Naval Medical University, Shanghai, China. [2]Jinling Hospital, Affiliated Hospital of Medical School, Nanjing University, Nanjing, China. [3]Department of Vascular and Endovascular Surgery, Changzheng Hospital, Naval Medical University/Second Military Medical University, Shanghai, China. [4]Department of Pharmacy, Shanghai Tenth People's Hospital, Tongji University School of Medicine, Shanghai, China. [5]Department of Diving and Hyperbaric Medicine, Naval Special Medical Center, Naval Medical University/Second Military Medical University, Shanghai, China. [6]Department of Cardiology, School of Medicine, Shanghai Tenth People's Hospital, Tongji University School of Medicine, Shanghai, China. [7]Shanghai Key Laboratory of Organ Transplantation, Fudan University, Shanghai, China. [8]Institute of Clinical Science, Zhongshan Hospital Fudan University, Shanghai, China. [9]Department of Orthopedic Surgery/Spine Center, Changzheng Hospital Affiliated Hospital of Naval Medical University/Second Military Medical University, Shanghai, China. [10]The National Demonstration Center for Experimental Pharmaceutical Education, Naval Medical University/Second Military Medical University, Shanghai, China. [11]These authors contributed equally: Di-Yang Sun, Wen-Bin Wu, Jian-Jin Wu, Yu Shi, Jia-Jun Xu. ✉e-mail: qulefeng@smmu.edu.cn; fumingshen@tongji.edu.cn; djli@tongji.edu.cn; pwang@smmu.edu.cn

