## [Peer Review File · Nature Communications]

Pro-ferroptotic signaling promotes arterial aging via vascular smooth muscle cell senescenceEditorial Note: This manuscript has been previously reviewed at another journal that is not operating a transparent peer review scheme. This document only contains reviewer comments and rebuttal letters for versions considered at *Nature Communications*. Mentions of prior referee reports have been redacted.

REVIEWER COMMENTS

Reviewer #1 (Remarks to the Author):

The authors have satisfied the critiques (reviewer one critiques).

Reviewer #4 (Remarks to the Author):

In reviewing the manuscript, I raise two fundamental concerns: 1) insufficient rigour in characterising ferroptosis and senescence and 2) over-interpretation of the data, which currently insufficiently evidences the authors conclusions.

Fundamental Issues

1. Insufficient rigour. The authors use different 'markers' as read-outs for ferroptosis throughout the manuscript, including C11-BODIPY oxidation/intracellular Fe²⁺ by FerroOrange/GSH to GSSG ratio. Whilst these particular assays are (at least in combination) ferroptosis-specific (as described in the Stockwell 2018 article cited by the authors (response to the reviewers, p42)), the authors also present assays that are not specific to ferroptosis, such as bulk NAD⁺ level/expression level of certain proteins (e.g. ACSL4, GPX4, CD38 and PARP1). Crucially, in some cases the authors only present data for these 'non-specific' ferroptosis assays, which are not (in isolation) robust indicators of ferroptosis. Senescence is similarly inadequately measured, mostly by SA-beta-gal staining intensity (when % SA-beta-gal positivity is the more useful metric - see below), and/or by p16 and p21 expression (both not specific to senescence). Senescence is a state of permanent cell cycle arrest and, since there are no completely specific markers for senescence, it is also necessary to functionally assess proliferation (e.g BrDu uptake) and cell death, and use a combination of these metrics to establish/quantify cells as senescent. Without rigorous and consistent methodology, the data presented are not convincing. For example, in the aged mouse model, senescence is assessed only by bulk p16 expression, whilst ferroptosis is only characterised by bulk NAD⁺ and GSH/GSSG (figure 4), why were other methodologies (used elsewhere) not applied to this (and other) model(s)?

2. Over-interpretation of the data. This is in part a product of a lack of thorough methodology as described above, however there is also a fundamental discrepancy in the definition of, and how the authors interpret, these processes. The assessments made for both ferroptosis and senescence was typically made on bulk cell populations, precluding (or at least data is not presented for) interpretation of the % of cells that are undergoing these processes, and there is no attempt to look at whether the same cells are expressing markers of these processes. One would expect heterogeneity in the cell populations, and though it is not possible to interpret, from the (intensity) data presented, the extent of cells undergoing these processes, in several experiments it appears that almost all cells within the population are 'senescent' (SA-beta-gal positive) and under the same conditions almost all cells within the population are also 'ferroptotic' (C11-BODIPY and/or FerroOrange positive). Examples include: replicatively senescent VSMC (figure 1), senescence induced (bleomycin treated) MOVAS cells (supp fig

1) and ferroptosis induced (RSL3 treated) VSMCs (figure 5). If ferroptosis is functionally defined as non-apoptotic iron-dependent cell death and senescence is defined as a state of permanent cell cycle arrest, which are by definition mutually exclusive within the same cell at a given time, how do the authors reconcile these data?

[REDACTED]

Point-to-point response to Reviewer #1

Remarks to the Author

The authors have satisfied the critiques (reviewer one critiques).

Response: We are grateful for your positive feedback regarding our revision and appreciate your acknowledgment of our dedicated efforts.

Point-to-point response to Reviewer #4

In reviewing the manuscript, I raise two fundamental concerns: 1) insufficient rigour in characterising ferroptosis and senescence and 2) over-interpretation of the data, which currently insufficiently evidences the authors conclusions.

Response: We express our heartfelt gratitude for your valuable feedback, which was insightful, professional, and constructive. Your unbiased assessment of the issue related to “vascular aging and atherosclerosis” in our previous manuscript is truly appreciated. After carefully reading and comprehending your comments, we fully agree with your concerns and have made several revisions to address them. These include: (1) conducting additional experiments; (2) rearranging the manuscript with improved narrative flow and logical presentation order; and (3) modifying the title and text to better reflect our findings. We will provide detailed explanations of these changes in our specific responses below.

Fundamental Issues

Question 1. Insufficient rigour. The authors use different 'markers' as read-outs for ferroptosis throughout the manuscript, including C11-BODIPY oxidation/intracellular Fe²⁺ by FerroOrange/GSH to GSSG ratio. Whilst these particular assays are (at least in combination) ferroptosis-specific (as described in the Stockwell 2018 article cited by the authors (response to the reviewers, p42)), the authors also present assays that are not specific to ferroptosis, such as bulk NAD⁺ level/expression level of certain proteins (e.g. ACSL4, GPX4, CD38 and PARP1). Crucially, in some cases the authors only present data for these 'non-specific' ferroptosis assays, which are not (in isolation) robust indicators of ferroptosis. Senescence is similarly inadequately measured, mostly by SA-beta-gal staining intensity (when % SA-beta-gal positivity is the more useful metric - see below), and/or by p16 and p21 expression (both not specific to senescence). Senescence is a state of permanent cell cycle arrest and, since there are no completely specific markers for senescence, it is also necessary to functionally assess proliferation (e.g. BrDu uptake) and cell death, and use a combination of these metrics

to establish/quantify cells as senescent. Without rigorous and consistent methodology, the data presented are not convincing. For example, in the aged mouse model, senescence is assessed only by bulk p16 expression, whilst ferroptosis is only characterised by bulk NAD⁺ and GSH/GSSG (figure 4), why were other methodologies (used elsewhere) not applied to this (and other) model(s)?

Response: We believe that the comment raised an important issue regarding the appropriateness, sufficiency, and consistency of some of the methods used in our manuscript. We value this feedback and have revised our manuscript from three aspects:

Concern 1: Methodological Inappropriateness. The referee pointed out that “*whilst these particular assays are (at least in combination) ferroptosis-specific (as described in the Stockwell 2018 article cited by the authors), the authors also present assays that are not specific to ferroptosis, such as bulk NAD⁺ level/expression level of certain proteins (e.g. ACSL4, GPX4, CD38 and PARP1)*”. We apologize for any confusion caused by our unclear expression. Please see the summarized image below for the read-outs in our study. The NAD⁺ level of bulk tissue or VSMCs, as well as protein expression of CD38 and PARP1 (two NAD⁺ consumers), were not used as ferroptosis markers. They are read-outs for vascular aging (see pink horizontal line). It is well accepted that NAD⁺ decline is a characteristic feature of aging in various organs/tissues.^{1,2} Our group previously reported hepatic NAD⁺ decline in aging. Previous investigations have found that senescent cells promote tissue NAD⁺ decline during ageing,³ and NAD⁺ supplementation is able to reduce cell senescence and slow aging process.⁴⁻⁶ Our group also reported that the VSMCs senescence could be attenuated by raising NAD⁺ or NAD⁺-dependent SIRT6/SIRT3 activity.^{7,8} Thus, our present work seeks to demonstrate that under vascular stress, the activation of ferroptosis or ferroptosis-related signals can induce the senescence of VSMCs, leading to arterial NAD⁺ loss, remodeling, and aging.

Please consult the summarized image for a graphical representation of the outcomes in our study. Cysteine/GSH/GPX4 constitutes the principal axis of the ferroptosis theory, with the GSH/GSSG ratio recognized as a pivotal metric for assessing ferroptosis, as per the guidelines in ferroptosis research.⁹ In contrast, NAD⁺ contents is not been considered as a specific marker of ferroptosis.

A summary of read-outs in our study

Regrettably, in response to the requirements of certain reviewers during previous revision, we juxtaposed the results of the GSH/GSSG ratio with NAD⁺ levels. This decision was influenced by the perception that GSH is intricately connected to NADPH⁺,¹⁰ thus implicating NAD⁺. Unfortunately, this arrangement inadvertently engendered logical confusion, creating the false impression that NAD⁺ is also a marker of ferroptosis. We sincerely apologize for any misunderstanding arising from this unintentional misalignment on our part.

Presently, we have integrated the results of GSH and the GSH/GSSG ratio alongside other established ferroptosis markers, including the labile Fe²⁺ pool measured through the FerroOrange probe. In a parallel manner, the protein expression levels of ACSL4 and GPX4, representing pro-ferroptosis and anti-ferroptosis factors, respectively, have been utilized as indicators of ferroptosis. In contrast, the NAD⁺ level and the protein expression of NAD⁺ consumers CD38 and PARP1 have been employed as read-outs to assess the tissue/cell NAD⁺ status. This comprehensive approach serves as an index for aging, acknowledging the role of NAD⁺ as a significant factor in the aging process.

We have included both the **original Figure 1** and the **re-organized Figure 1**, along with the **original Figure S1** and **re-organized Figure 1**, to provide a comprehensive explanation. Notably, we have segregated the results of NAD⁺ and the GSH/GSSG ratio. This reorganization extends to other figures, such as **Figure S2, Figure S3, Figure S4, Figure 2, Figure 3, Figure S7, and Figure 4**. The relevant results have correspondingly been amended at specific points in the manuscript (**Line 145-146, 221, 229, 277, and 309**). We are confident that these revisions enhance the clarity of our manuscript for readers. We appreciate your insightful feedback, which has contributed to the professional refinement of our work.

Revision of Figure 1

Please note the contents of NAD⁺ and GSH/GSSG ratio were segregated (green box)

Revision of Figure S1

Please note the contents of NAD⁺ and GSH/GSSG ratio were segregated (green box)

A SA-β-gal staining

Relative SA-β-gal staining intensity (fold)

B

γH2A.X

DAPI

Merge

NAD⁺ level

C

Relative NAD⁺ level (μmol/g protein)

Control Ang II + Bleo

D

ACSL4 70

ALOX15 75

TFR1 100

DMT1 60

IRP2 72

GPX4 23

Tubulin 53

Control Ang II + Bleo

Relative protein (fold)

ACSL4

ALOX15

TFR1

DMT1

IRP2

GPX4

E FerroOrange/Hoechst

Relative labile ferrous iron contents (fold)

Control Ang II + Bleo

F

GSH/GSSG ratio

Control Ang II + Bleo

GSH/GSSG ratio

Concern 2: Methodological Appropriateness. The referee pointed out that “*Senescence is a state of permanent cell cycle arrest and, since there are no completely specific markers for senescence, it is also necessary to functionally assess proliferation (e.g BrdU uptake) and cell death, and use a combination of these metrics to establish/quantify cells as senescent*”. We wholeheartedly concur with this perspective. As outlined in the summary of our study's read-outs, we employed a comprehensive approach, incorporating SA- β -gal staining, senescence markers (immunoblotting or immunohistochemistry of γ H2A.X, p16INK4A, p21WAF1, and p53), and the expression of SASP factors (IL-1 α and HMGB1). In response to your valuable comment, we have introduced additional experiments involving Click-iT® EdU labeling and cell viability assessments in our cell culture studies (**Figure S1C, Figure S1D, Figure S4C, Figure S4D, Figure S7B, Figure S7C, Figure S10A, and Figure S10B**). The results and methods were also incorporated into the revised manuscript (**Line 143, 180, 245, 311, 337, and 874-878**). The results from these new experiments underscore that the VSMCs characterized as 'senescent' exhibit notable reductions in EdU incorporation and cell viability. This supplementary evidence further supports the assertion that the VSMCs in our study indeed manifest senescent characteristics.

Concern 3: Methodological Inconsistency. The referee pointed out that “*For example, in the aged mouse model, senescence is assessed only by bulk p16 expression, whilst ferroptosis is only characterised by bulk NAD⁺ and GSH/GSSG (figure 4), why were other methodologies (used elsewhere) not applied to this (and other) model(s)?*”. We acknowledge this inconsistency, particularly in the presentation of p16 expression for senescence and bulk NAD⁺ and GSH/GSSG for ferroptosis in the previous **Figure 4**.

In response to your valuable feedback, we have taken substantial measures in this revision to rectify this inconsistency. The previous **Figure 4A, 4B, 4C, and 4I** have been relocated to the Supplemental data as **Supplemental Figure 8A, 8B, 8C, and 8F**. To address the concern about the inducible vascular aging model (caused by Ang II+Bleo infusion), we have introduced **new Figure 4D** (depicting p21 and γ H2A.X protein expression), **Supplemental Figure 8D** (illustrating ACSL4 and TFR1 protein expression), and **Supplemental Figure 8E** (highlighting MDA levels). Additionally, to showcase the anti-senescence effects of AAV carrying wild-type GPX4 versus mutant GPX4 in the natural vascular aging model (old mice), we have included **new Figure 4H** (depicting ACSL4 protein expression), **new Figure 4I** (illustrating MDA levels), and **new Figure 4K** (showcasing p21 and γ H2A.X protein expression). The presentation order of these results aligns with the earlier figure summarizing our read-outs: 1) markers of ferroptosis; 2) markers of senescence; 3) markers of vascular

remodeling and aging.

We trust that these revisions comprehensively address your concerns, and we express our gratitude for your valuable and professional comments.

Question 2. Over-interpretation of the data. This is in part a product of a lack of thorough methodology as described above, however there is also a fundamental discrepancy in the definition of, and how the authors interpret, these processes. The assessments made for both ferroptosis and senescence was typically made on bulk cell populations, precluding (or at least data is not presented for) interpretation of the % of cells that are undergoing these processes, and there is no attempt to look at whether the same cells are expressing markers of these processes. One would expect heterogeneity in the cell populations, and though it is not possible to interpret, from the (intensity) data presented, the extent of cells undergoing these processes, in several experiments it appears that almost all cells within the population are 'senescent' (SA-beta-gal positive) and under the same conditions almost all cells within the population are also 'ferroptotic' (C11-BODIPY and/or FerroOrange positive). Examples include: replicatively senescent VSMC (figure 1), senescence induced (bleomycin treated) MOVAS cells (supp fig 1) and ferroptosis induced (RSL3 treated) VSMCs (figure 5). If ferroptosis is functionally defined as non-apoptotic iron-dependent cell death and senescence is defined as a state of permanent cell cycle arrest, which are by definition mutually exclusive within the same cell at a given time, how do the authors reconcile these data?

Response: After careful consideration of your insightful comment, we have undertaken a thorough reevaluation of our manuscript, and we wish to elucidate our reflections and modifications as follows:

Concern 1: Heterogeneity of cellular senescence. We fully acknowledge the significance of the heterogeneity of cellular senescence, as highlighted in your comment. In our study, we observed heterogeneity in cellular senescence, evident from the variability in SA- β -gal staining positivity among VSMCs treated with Ang II+Bleo, as depicted in the right image. Moreover, we concur with your observation that

both ferroptosis and senescence were primarily evaluated in bulk cell populations in our work. Our research methods encompassed SA- β -gal staining and the assessment of expression levels of γ H2A.X, p16, and p21—widely recognized markers for senescent cells at the bulk

transcript level. While these approaches provide valuable insights, we acknowledge the emergence of advanced techniques, particularly single-cell RNA sequencing (scRNA-seq), as powerful tools for investigating the heterogeneity of cellular senescence. For instance, Saul et al. introduced the 'SenMayo' gene set, utilizing scRNA-seq to characterize senescent cells at the single-cell level and identify key intercellular signaling pathways (*Nat Commun* 2022).¹¹ Similarly, Zhang et al., employing scRNA-seq, identified a subpopulation of old fibroadipogenic progenitors expressing p16^{Ink4a} and other senescence-related genes, demonstrating concurrent DNA damage and chromatin reorganization (*Nat Aging* 2022).¹² Recognizing the limitation of not conducting a single-cell level analysis in assessing the heterogeneity of cellular senescence in our study, we acknowledge the need for further investigation. We have incorporated the above sentiments as a dedicated paragraph in the 'Discussion' section of the manuscript to openly address this limitation (**Line 624-631**). We appreciate your valuable feedback, which has prompted a constructive discussion of this aspect of our work.

Concern 2: Co-existence of ferroptosis and senescence in cells. Our work is not the first study to show the co-existence of ferroptosis with senescence in the cells. Sun et al. showed that glutathione depletion induces ferroptosis and senescence in retinal pigment epithelial cells.¹³ Li et al. found 1-methyl-4-phenylpyridinium induces ferroptosis and senescence in PC12 cells.¹⁴ Xu et al. demonstrated that vitamin D receptor attenuates osteoblastic ferroptosis and senescence.¹⁵ Li et al. reported that particulate matter 2.5 could trigger pulmonary inflammation and fibrosis by inducing pulmonary epithelial senescence and ferroptosis.¹⁶ However, these results merely demonstrated the co-existence of ferroptosis and senescence without elucidating their interrelationship. It is important to note that none of the aforementioned studies provided direct evidence for the co-existence of ferroptosis and senescence in cells.

In our manuscript, we endeavored to address the crucial question of "in ferroptosis and senescence, which event is the earlier event, and which one is the initiating factor?" Through conventional medium transfer experiments, we observed that the culture medium from ferroptotic VSMCs (where not all VSMCs perished in the culture dish) exerted a potent promotional effect on senescence in normal VSMCs cultured separately, while the reverse scenario did not hold true (**Figure 5G-L**). We noted that the transferred culture medium from VSMCs underwent apoptosis (induced by ABT-737) and necroptosis (induced by TNF- α , SM-164 and z-VAD-fmk, referred as TSZ) failed to induce senescence (**Figure 5I**). At the time of this observation (around 2021), these findings were met with excitement. We also observed parallels with the work of Nishizawa *et al.*, who reported that lipid peroxidation and its

consequences propagate from ferroptotic cells to surrounding cells, even in the absence of direct exposure to a ferroptosis inducer¹⁷. They concluded that ferroptotic cells function not only as dying cells but also as signal transmitters, initiating a cascade of further ferroptosis. Our results align with their observations, suggesting that ferroptotic cells may act as signal transmitters inducing senescence both within themselves and in surrounding cells.

In response to your concern about whether the same cells express markers of these processes, we conducted additional experiments in this round of revision. We investigated the effects of RSL3, a well-established ferroptosis activator, at a non-cytotoxic dose (0.5 μ M) on cell viability, GSH/GSSG ratio, and EdU incorporation. Treatment with RSL3 at 0.5 μ M for 24 hours did not induce cell death, as indicated by the gradual inhibition of cell viability observed after 3-5 days (**new Figure 5A**). However, an evident decline in the GSH/GSSG ratio was observed after 24 hours, suggesting the activation of pro-ferroptosis signaling pathways in the cells (**new Figure 5B**). Inhibition of EdU incorporation was observed on Day 3 and Day 5 but not on Day 1, implying that ferroptosis activation appears to precede senescence (**new Figure 5B**). Similar results were observed in cell viability (**new Figure 5C**). These results provide further insights into the temporal sequence of ferroptosis and senescence activation.

New Figure 5A-C

To provide a more comprehensive response to your query, we conducted preliminary experiments using a combination of a 'ferroptosis probe' (Liperfluo probe¹⁸, green) and a 'senescence probe' (KSAP1, red)¹⁹ to stain VSMCs treated with RSL3, a ferroptosis activator, at a non-cytotoxic dose (0.5 μ M) for 3 days. As illustrated in the **new Figure 5F** as below, the green signal (Liperfluo dye) was observable in nearly all VSMCs, indicating the activation of ferroptosis stress in these cells. In contrast, the red signal of the senescence probe (KSAP1) was only evident in select VSMCs (highlighted by pink arrows) and not in all VSMCs. Notably, the 'non-senescent' VSMCs with minimal red signal (indicated by white arrows) maintained a spindle-shaped appearance, while the 'senescent' VSMCs with a noticeable red signal exhibited a flat and enlarged morphology, which aligns with the recognized characteristic appearance of senescent VSMCs²⁰.

New Figure 5F

These results may address your concern on the existence of ferroptosis and senescence within the same cells. Moreover, these results indicate that the activation of ferroptosis signaling pathway may be earlier than senescence in the RSL3-treated VSMCs. When considering these findings alongside our earlier results demonstrating the inhibition of ferroptosis through a chemical inhibitor or knockin of GPX4 (**Figure 2, 3 and 4**), our collective observations lend support to the conclusion that the activation of the ferroptosis signaling pathway could potentially induce senescence in stressed VSMCs.

Concern 3: The possible inherent contradictions between ferroptosis and senescence. You raised a concern that “*If ferroptosis is functionally defined as non-apoptotic iron-dependent cell death and senescence is defined as a state of permanent cell cycle arrest, which are by definition mutually exclusive within the same cell at a given time, how do the authors reconcile these data?*” Your concern about potential inherent contradictions between ferroptosis and senescence has been duly noted. The apparent contradiction arises from the functional definitions, where ferroptosis is characterized as non-apoptotic iron-dependent cell death, and senescence is described as a state of permanent cell cycle arrest, which, by definition, are considered mutually exclusive within the same cell at a given time. We acknowledge this concern and recognize that, as shown in our study, both 'ferroptosis' and 'senescence' markers have been observed in the same cells, indicating the coexistence of these processes. It's essential to clarify that, per their definitions, 'ferroptosis' implies cell death, whereas 'senescence' denotes a cell cycle arrest without death. Numerous studies have illustrated that the activation of pro-ferroptotic signaling factors does not necessarily lead to irreversible cell death, especially in cardio-cerebral-renal vasculature. Ferroptotic stress has been reported to induce a transient inflammatory state in proximal tubular cells (*Elife* 2021),²¹ increase cyst growth(*J Am Soc Nephrol* 2021),²² and drive cellular calcification (*Kidney international* 2022),²³ in addition to triggering cell death.

Moreover, the understanding of senescence is evolving. The p16^{INK4A+} senescent cells in the basement membrane form a reparative niche in the lung.²⁴ Senescent cells hyperactivates hair growth by enhancing the activity of adjacent intact stem cells.²⁵ Stress-induced pancreatic β cell early senescence confers protection against type 1 diabetes.²⁶ Specially, senescent cells promote newt limb regeneration by promoting muscle dedifferentiation.²⁷ Considering our recent findings of activated ferroptotic stress facilitating smooth muscle cell dedifferentiation in injured arteries²⁸, senescence may act as a link between ferroptosis and dedifferentiation/remodeling of the vascular wall.

Taking these considerations into account, we deemed the term 'ferroptosis,' implying cell death, might not precisely reflect our major findings. Therefore, we have modified the title to "*Ferroptosis **stress** drives vascular smooth muscle cell senescence to promote arterial NAD⁺ loss, remodeling, and aging.*" This adjustment aims to distinguish between the stress related to ferroptosis and the actual cell death process. Similar changes have been made in the abstract (**Line 52 and 61**) and throughout the manuscript to maintain clarity in conveying our findings.

[REDACTED]

References

1. Camacho-Pereira, J., *et al.* CD38 Dictates Age-Related NAD Decline and Mitochondrial Dysfunction through an SIRT3-Dependent Mechanism. *Cell Metab* **23**, 1127-1139 (2016).
2. Tarrago, M.G., *et al.* A Potent and Specific CD38 Inhibitor Ameliorates Age-Related Metabolic Dysfunction by Reversing Tissue NAD(+) Decline. *Cell Metab* **27**, 1081-1095 e1010 (2018).
3. Covarrubias, A.J., *et al.* Senescent cells promote tissue NAD(+) decline during ageing via the activation of CD38(+) macrophages. *Nat Metab* **2**, 1265-1283 (2020).
4. Hou, Y., *et al.* NAD(+) supplementation reduces neuroinflammation and cell senescence in a transgenic mouse model of Alzheimer's disease via cGAS-STING. *Proc Natl Acad Sci U S A* **118**(2021).
5. Zhan, R., *et al.* NAD(+) rescues aging-induced blood-brain barrier damage via the CX43-PARP1 axis. *Neuron* (2023).
6. Zhang, H., *et al.* NAD(+) repletion improves mitochondrial and stem cell function and enhances life span in mice. *Science* **352**, 1436-1443 (2016).
7. Chi, C., *et al.* Exerkine fibronectin type-III domain-containing protein 5/irisin-enriched extracellular vesicles delay vascular ageing by increasing SIRT6 stability. *Eur Heart J* **43**, 4579-4595 (2022).
8. Li, D.J., *et al.* alpha7 Nicotinic Acetylcholine Receptor Relieves Angiotensin II-Induced Senescence in Vascular Smooth Muscle Cells by Raising Nicotinamide Adenine Dinucleotide-Dependent SIRT1 Activity. *Arterioscler Thromb Vasc Biol* **36**, 1566-1576 (2016).
9. Hadian, K. & Stockwell, B.R. SnapShot: Ferroptosis. *Cell* **181**, 1188-1188 e1181 (2020).
10. Moreno-Sanchez, R., *et al.* Control of the NADPH supply and GSH recycling for oxidative stress management in hepatoma and liver mitochondria. *Biochim Biophys Acta Bioenerg* **1859**, 1138-1150 (2018).
11. Saul, D., *et al.* A new gene set identifies senescent cells and predicts senescence-associated pathways across tissues. *Nat Commun* **13**, 4827 (2022).
12. Zhang, X., *et al.* Characterization of cellular senescence in aging skeletal muscle. *Nat Aging* **2**, 601-615 (2022).
13. Sun, Y., Zheng, Y., Wang, C. & Liu, Y. Glutathione depletion induces ferroptosis, autophagy, and premature cell senescence in retinal pigment epithelial cells. *Cell death & disease* **9**, 753 (2018).
14. Li, S., *et al.* p53-mediated ferroptosis is required for 1-methyl-4-phenylpyridinium-induced senescence of PC12 cells. *Toxicology in vitro : an international journal published in association with BIBRA* **73**, 105146 (2021).
15. Xu, P., *et al.* VDR activation attenuates osteoblastic ferroptosis and senescence by stimulating the Nrf2/GPX4 pathway in age-related osteoporosis. *Free radical biology & medicine* **193**, 720-735 (2022).
16. Li, N., Xiong, R., Li, G., Wang, B. & Geng, Q. PM2.5 contributed to pulmonary epithelial senescence and ferroptosis by regulating USP3-SIRT3-P53 axis. *Free radical biology & medicine* **205**, 291-304 (2023).
17. Nishizawa, H., *et al.* Lipid peroxidation and the subsequent cell death transmitting from

- ferroptotic cells to neighboring cells. *Cell death & disease* **12**, 332 (2021).
18. Wang, W., *et al.* CD8(+) T cells regulate tumour ferroptosis during cancer immunotherapy. *Nature* **569**, 270-274 (2019).
 19. Gao, Y., *et al.* Two-Dimensional Design Strategy to Construct Smart Fluorescent Probes for the Precise Tracking of Senescence. *Angew Chem Int Ed Engl* **60**, 10756-10765 (2021).
 20. Gardner, S.E., Humphry, M., Bennett, M.R. & Clarke, M.C. Senescent vascular smooth muscle cells drive inflammation through an interleukin-1 α -dependent senescence-associated secretory phenotype. *Arteriosclerosis, thrombosis, and vascular biology* **35**, 1963-1974 (2015).
 21. Ide, S., *et al.* Ferroptotic stress promotes the accumulation of pro-inflammatory proximal tubular cells in maladaptive renal repair. *Elife* **10**(2021).
 22. Zhang, X., *et al.* Ferroptosis Promotes Cyst Growth in Autosomal Dominant Polycystic Kidney Disease Mouse Models. *J Am Soc Nephrol* **32**, 2759-2776 (2021).
 23. Ye, Y., *et al.* Repression of the antiporter SLC7A11/glutathione/glutathione peroxidase 4 axis drives ferroptosis of vascular smooth muscle cells to facilitate vascular calcification. *Kidney Int* **102**, 1259-1275 (2022).
 24. Reyes, N.S., *et al.* Sentinel p16(INK4a+) cells in the basement membrane form a reparative niche in the lung. *Science* **378**, 192-201 (2022).
 25. Wang, X., *et al.* Signalling by senescent melanocytes hyperactivates hair growth. *Nature* **618**, 808-817 (2023).
 26. Lee, H., *et al.* Stress-induced beta cell early senescence confers protection against type 1 diabetes. *Cell Metab* (2023).
 27. Walters, H.E., Troyanovskiy, K.E., Graf, A.M. & Yun, M.H. Senescent cells enhance newt limb regeneration by promoting muscle dedifferentiation. *Aging Cell* **22**, e13826 (2023).
 28. Ji, Q.X., *et al.* Ferroptotic stress facilitates smooth muscle cell dedifferentiation in arterial remodelling by disrupting mitochondrial homeostasis. *Cell Death Differ* **30**, 457-474 (2023).

REVIEWER COMMENTS

Reviewer #4 (Remarks to the Author):

In this revision, the Authors have substantially improved the content and presentation of the manuscript. The addition of functional assessments for senescence (cell proliferation by EdU incorporation and cell viability) alongside more comprehensive assessments of NAD⁺ status and pro-ferroptotic signalling and attempts to investigate these processes simultaneously within the same cells, have added weight to the previous data, such that the findings are now convincing. The authors have also made substantial changes to the title and presentation of the manuscript to clarify the key concepts and to remove ambiguity that is inherent in the functional definitions of ferroptosis and senescence.

There are numerous instances where language needs revising for grammatical accuracy and comprehension, I've only highlighted instances where meaning is adversely affected.

Minor comments:

Whilst most of the experiments described in the manuscript have been strengthened by more rigorous assessment of senescence and pro-ferroptotic signalling, there remain a couple of instances of over-interpretation of a single end-point or read-out:

p12 line 314 - please change 'induction of senescence' to 'effects on cell proliferation and death' or similar. It is not appropriate to conclude anything other than this based on the EdU incorporation and viability data presented in fig 5A-C

p15 line 412, please change 'rosiglitazone inhibited senescence' to 'rosiglitazone reduced H2AX staining' or similar; senescence per se has not been addressed here

Whilst the authors have changed the title and explained very well in their rebuttal (under concern 3: the possible inherent contradictions between ferroptosis and senescence), that their interpretation is that (potentially sub-lethal) pro-ferroptotic signalling is linked to senescence, I believe this has still not been adequately explained or addressed in the manuscript - for example, the term ferroptosis is still used (with no clarification or reference to ferroptosis-stress or pro-ferroptotic signalling) throughout the discussion. For clarity and consistency, this should be addressed throughout the manuscript.

For balance, the discussion would benefit from inclusion of a small paragraph or sentences describing other potential mechanisms by which aging and, importantly (given the dependence of most of the experimental data contained in the manuscript on this experimental model), on other (published) effects of angiotensin and/or bleomycin treatment that can induce senescence independently of pro-ferroptotic

signalling, such as genomic DNA damage and mitochondrial damage. Although the authors build a strong case, using chemical and genetic manipulation of pro-ferroptotic signalling specific agonists/inhibitors, that pro-ferroptotic signalling can lead to senescence, other (ferroptosis-independent) effects are likely to contribute and this is relevant in vivo and with respect to therapeutic targeting. It is also important to acknowledge that, during the revision of this manuscript, it has recently been published that sub-lethal pro-apoptotic signalling can lead to senescence (e.g. Victorelli et al, Nature 2023).

Further minor changes:

p7 lines 179-181, currently reads that liproxstatin-1 reduced cell proliferation (EdU incorporation) and viability - it did not! The authors likely mean that liproxstatin-1 ameliorated or reduced the effects of angII + bleo on these parameters - please correct

p7 line 182, please change Supplemental Figure 4E to 4G

p7 line 184, please change Supplemental Figure 4F to 4H

p7 line 187, please change Supplemental Figure 4G-H to 4E-F

p9 line 228, 'disruption of ferroptosis genetic inhibition on cellular senescence' has no sense - please clarify

p10, please change all cases of 'VSMC-specifically' to 'VSMC-specific'

p11 line 286, please change 'Genetically' to 'Genetic'

p15 line 409, please change Supplemental Figure 10C to 11C

Supplementary Figure 3 legend and Figures B and D, incorrect labelling/graphs - the legend and graph axis labels are not consistent in B and D. Also it appears that GSH level not the GSH/GSSG ratio (as stated in the manuscript text (p6 line 170)) has been presented

Supplementary Figure 10, please check the data, axis labels and legends - the images in D and F are labelled differently (+NMN) to the accompanying graphs (+NAC); the data in E (labelled delayed NAC) also looks the same as the data in the previous Supplementary Figure 9E which was previously labelled +liproxstatin-1

Supplementary Figure 11 - the newly supplied blot (and corresponding quantification) is undoubtedly more convincing and more consistent with the interpretation of the authors. However, how was this blot generated? From new samples? The adjacent blots (for NCOA4 and loading control H2B) have not changed/been updated??

Point-to-point response to Reviewer #4

Remarks: In this revision, the Authors have substantially improved the content and presentation of the manuscript. The addition of functional assessments for senescence (cell proliferation by EdU incorporation and cell viability) alongside more comprehensive assessments of NAD⁺ status and pro-ferroptotic signaling and attempts to investigate these processes simultaneously within the same cells, have added weight to the previous data, such that the findings are now convincing. The authors have also made substantial changes to the title and presentation of the manuscript to clarify the key concepts and to remove ambiguity that is inherent in the functional definitions of ferroptosis and senescence. There are numerous instances where language needs revising for grammatical accuracy and comprehension, I've only highlighted instances where meaning is adversely affected.

Response: We appreciate your professional and constructive feedbacks. In this revision, we systematically addressed the issues you brought to our attention, meticulously revising our manuscript based on your comments by refraining from over-interpreting the results. We are confident that we have effectively resolved all your concerns. Thank you sincerely for your comprehensive review and valuable input on our manuscript.

Minor comments: Whilst most of the experiments described in the manuscript have been strengthened by more rigorous assessment of senescence and pro-ferroptotic signalling, there remain a couple of instances of over-interpretation of a single end-point or read-out.

Response: Based on your feedback, we have moderated our statement and refined our interpretation. We sincerely appreciate your precise guidance on the relationship between senescence and pro-ferroptotic signaling.

Q1. p12 line 314 - please change 'induction of senescence' to 'effects on cell proliferation and death' or similar. It is not appropriate to conclude anything other than this based on the EdU incorporation and viability data presented in fig 5A-C.

Response: We have changed 'induction of senescence' to 'effects on cell proliferation and death' accordingly (**Line 316**).

314 in cell viability became apparent at 3 days post-incubation (**Figure 5B-C**). These findings
315 suggest that the ferroptosis inducer RSL3 promptly induced **ferroptosis stress**, appearing to
316 precede its **effects on cell proliferation and death**. At 5 days post-incubation, RSL3 at a non-
317 cytotoxic dose (0.5 μ M) significantly increased SA- β -gal activity (**Figure 5D**), and upregulated
318 p16^{INK4A} and p21^{WAF1} (**Figure 5E**), suggesting an induction of senescence by **pro-ferroptosis**

Q2. p15 line 412, please change 'rosiglitazone inhibited senescence' to 'rosiglitazone reduced H2AX staining' or similar; senescence per se has not been addressed here.

Response: We completely agree with you that γ H2A.X staining alone cannot accurately reflect senescence. It was an over-interpretation. In response to your comment, we have revised 'rosiglitazone inhibited senescence' to 'rosiglitazone reduced γ H2AX staining' (**Line 414**) Thank you.

413 further aggravated senescence-associated lipid peroxidation (**Figure 6K**). Conversely,
414 activation of PPAR γ using **rosiglitazone reduced γ H2AX staining** (**Figure 6L**). These results
415 indicate that inhibition of **ferroptosis signaling** may suppress ferritinophagy and affect non-
416 nuclear PPAR γ signalling in vascular senescence.↵

Q3. Whilst the authors have changed the title and explained very well in their rebuttal (under concern 3: the possible inherent contradictions between ferroptosis and senescence), that their interpretation is that (potentially sub-lethal) pro-ferroptotic signaling is linked to senescence, I believe this has still not been adequately explained or addressed in the manuscript - for example, the term ferroptosis is still used (with no clarification or reference to ferroptosis-stress or pro-ferroptotic signaling) throughout the discussion. For clarity and consistency, this should be addressed throughout the manuscript.

Response: We greatly appreciate your attention to the incomplete amendment of the term 'ferroptosis,' which may not fully address the inherent contradictions between ferroptosis and senescence. To rectify this, we have replaced the term 'ferroptosis' with 'pro-ferroptotic signaling', 'ferroptosis stress', or 'ferroptosis activation' depending on the context. In this revised version, all modified phrases have been highlighted in yellow throughout the manuscript, including in figure legends and supplemental figures. For example, we pasted the Abstract section as below for your reference. You can go through the whole manuscript and examine our revisions on this issue.

48 **Abstract**[¶]
49 Senescence of vascular smooth muscle cells (VSMCs) contributes to aging-related
50 cardiovascular diseases (CVDs) by promoting arterial remodelling and stiffness. Ferroptosis
51 is a novel type of regulated cell death associated with lipid oxidation. Here, we show that
52 **ferroptosis stress** drives VSMCs senescence to accelerate vascular NAD⁺ loss, remodelling
53 and aging. **Ferroptosis stress** is triggered in senescent VSMCs and arteries of aged mice.
54 Furthermore, **ferroptosis signaling activation** in VSMCs not only induces NAD⁺ loss and
55 senescence but also promotes the release of a pro-senescent secretome. Pharmacological or
56 genetic inhibition of **pro-ferroptosis signalling**, ameliorates VSMCs senescence, reduces
57 vascular stiffness and retards the progression of abdominal aortic aneurysm in mice.
58 Mechanistically, we revealed that inhibition of **ferroptosis signalling** facilitates the nuclear-
59 cytoplasmic shuttling of proliferator-activated receptor- γ (PPAR γ) and, thereby impeding
60 NCOA4-ferritin complex-centric ferritinophagy. Finally, the activated ferroptotic status
61 correlates with arterial stiffness in a human proof-of-concept study. These findings have
62 significant implications for future therapeutic strategies aiming to eliminate vascular **ferroptosis**
63 **stress** in senescence- or aging-associated CVDs.[¶]

Furthermore, we considered it essential to directly address this seemingly contradiction. So, we incorporated a paragraph raising the issue of potential inherent contradiction between ferroptosis and senescence, along with our reflections on this matter (**Lines 558-578**). While our proficiency in English may not be entirely sufficient to convey our thoughts comprehensively, we sincerely hope that our revised version meets your expectations.

Q4. For balance, the discussion would benefit from inclusion of a small paragraph or sentences describing other potential mechanisms by which aging and, importantly (given the dependence of most of the experimental data contained in the manuscript on this experimental model), on other (published) effects of angiotensin and/or bleomycin treatment that can induce senescence independently of pro-ferroptotic signalling, such as genomic DNA damage and mitochondrial damage. Although the authors build a strong case, using chemical and genetic manipulation of pro-ferroptotic signalling specific agonists/inhibitors, that pro-ferroptotic signalling can lead to senescence, other (ferroptosis-independent) effects are likely to contribute and this is relevant in vivo and with respect to therapeutic targeting. It is also important to acknowledge that, during the revision of this manuscript, it has recently been published that sub-lethal pro-apoptotic signalling can lead to senescence (e.g. Victorelli et al, *Nature* 2023).

Response: We totally agree with your comment that Ang II or bleomycin can induce senescence in a ferroptosis-independent manner. In response to your suggestion, we have added a brief paragraph describing the other potential mechanisms underlying the senescence-inducing action of Ang II or bleomycin in our models (**Lines 534-538**).

The important work you highlighted (Victorelli et al, *Nature* 2023)¹ suggest that pro-apoptotic stress causes mitochondrial DNA leakage to drive senescence. In fact, there is another study showing that pro-pyroptosis stress drives senescence (Fernández-Duran et al., *Cell Death Differ.* 2022;29:1267-1282).² We added a paragraph to discuss this interesting phenomenon (**Lines 558-578**). It appears that certain triggers induce cells toward senescence without reaching the threshold of cell death. In other words, there might be an interconversion between senescence and cell death in specific context. These two works were also cited. We greatly appreciate your constructive comment, which enhanced the quality of our manuscript.

Q5. p7 lines 179-181, currently reads that liproxstatin-1 reduced cell proliferation (EdU incorporation) and viability - it did not! The authors likely mean that liproxstatin-1 ameliorated or reduced the effects of angII + bleo on these parameters - please correct

Response: We apologize for the syntax error in this sentence. The paragraph was corrected to “*Liproxstatin-1 administration not only pronouncedly reduced SA-β-gal activity and γH2A.X level (Supplemental Figure 4A-B), but also ameliorated the inhibitory effects of Ang II+Bleo on EdU incorporation and cell viability (Supplemental Figure 4C-D)*” (Lines 181-182). Thank you.

Q6. p7 line 182, please change Supplemental Figure 4E to 4G; p7 line 184, please change Supplemental Figure 4F to 4H; p7 line 187, please change Supplemental Figure 4G-H to 4E-F/.

Response: We apologize for these errors. They have been corrected.

184 senescence markers (p53, p16^{INK4A} and p21^{WAF1}), as well as inhibited the release of SASP
185 factors interleukin-1α (IL-1α) and high mobility group box-1 protein (HMGB1) into culture
186 medium from senescent VSMCs (**Supplemental Figure 4E-F**). Intracellular NAD⁺ levels were
187 depleted in senescent VSMCs but partially rescued by liproxstatin-1 (**Supplemental Figure**
188 **4G**). Corresponding changes in NAD⁺ consumers PARP-1 and CD38 were observed
189 (**Supplemental Figure 4H**). ↵

Q7. p9 line 228, 'disruption of ferroptosis genetic inhibition on cellular senescence' has no sense - please clarify.

Response: The original sentences is ‘*We next assessed whether the disruption of ferroptosis genetic inhibition on cellular senescence would delay natural vascular aging*’, which was incorrect grammatically. In the revision, we changed it to ‘*We next assessed whether genetic inhibition of ferroptosis could postpone the natural aging process of blood vessels*’ (Line 230-231).

228 infusion (**Figure 3D**). These suggest knockin of GPX4 alleviates experimental vascular
229 senescence in adult mice. ↵
230 **We next assessed whether genetic inhibition of ferroptosis could postpone the natural**
231 **aging process of blood vessels.** The ferroptosis markers and GSH/GSSG ratio were reduced
232 and increased respectively in aortae of aged R26-GPX4 mice (24-month-old) compared with
233 those from aged WT mice (**Supplemental Figure 6C-D**). Fluorescent immunohistochemistry

Q8. p10, please change all cases of 'VSMC-specifically' to 'VSMC-specific'

Response: Corrected. Thank you.

262 vein and infused these mice with Ang II+Bleo (**Supplemental Figure 8A**). **VSMC-specific**
263 gene delivery by these AAV was confirmed using analyses of immunofluorescence
264 (**Supplemental Figure 8B**) and immunoblotting (**Supplemental Figure 8C**). **VSMC-specific**
265 overexpression of GPX4^{wt} blocked the Ang II+Bleo-induced alterations in GSH/GSSG ratio
266 (**Figure 4B**), ACSL4 and TFR1 protein expression (**Supplemental Figure 8D**), and MDA
267 content (**Supplemental Figure 8E**) in arterial tissue, confirming its inhibitory effect on
268 **ferroptosis stress**. Immunohistochemistry showed **VSMC-specific** overexpression of GPX4^{wt}
269 inhibited p16^{INK4A} accumulation in arterial wall (**Figure 4C**). Immunoblotting showed similar
270 changes in p21 and γ H2A.X (**Figure 4D**). Moreover, **VSMC-specific** overexpression of GPX4^{wt}

Q9. p11 line 286, please change 'Genetically' to 'Genetic'

Response: All the phrases '*Genetically inhibition of...*' in the manuscript were changed to '*Genetic inhibition of...*'. Thank you for your careful reading.

Q10. p15 line 409, please change Supplemental Figure 10C to 11C

Response: Corrected. I apologize for this oversight and appreciate your thorough review.

Q11. Supplementary Figure 3 legend and Figures B and D, incorrect labelling/graphs - the legend and graph axis labels are not consistent in B and D. Also it appears that GSH level not the GSH/GSSG ratio (as stated in the manuscript text (p6 line 170)) has been presented.

Response: Corrected. I apologize for this oversight and appreciate your thorough review.

63 **Supplemental Figure 3. Treatment of angiotensin II (Ang II) plus bleomycin (Bleo)**
64 **promotes ferroptosis stress, senescence and NAD⁺ loss in aged mice.**↵
65 **(A)** Fluorescent immunocytochemistry and quantitative analysis of ferroptosis marker ACSL4
66 in aortae of mouse infused with saline (Control) or Ang II+Bleo for 2 weeks. Nuclei were
67 stained by DAPI. ↵
68 **(B)** GSH levels in aortae of mouse infused with saline (Control) or Ang II+Bleo for 2 weeks. ↵
69 **(C)** Fluorescent immunocytochemistry and quantitative analysis of senescence marker H2A.X
70 in aortae of mouse infused with saline (Control) or Ang II+Bleo for 2 weeks. Nuclei were
71 stained by DAPI. H2A.X colocalizes with DAPI (nuclei). ↵
72 **(D)** NAD⁺ levels in aortae of mouse infused with saline (Control) or Ang II+Bleo for 2 weeks.↵
73 **(E)** Immunoblotting analysis of two NAD⁺ consumers CD38 and PARP-1 in aortae of mouse
74 infused with saline (Control) or Ang II+Bleo for 2 weeks.↵
75 Data expressed the mean±SEM. **P<0.01. Comparisons of parameters were performed with
76 Unpaired t-test. ↵

Q12. Supplementary Figure 10, please check the data, axis labels and legends - the images in D and F are labelled differently (+NMN) to the accompanying graphs (+NAC); the data in E (labelled delayed NAC) also looks the same as the data in the previous Supplementary Figure 9E which was previously labelled +lipoxstatin-1

Response: We meticulously reviewed Supplementary Figure 10 and discovered significant disarray in the labels, legends, and main-text description during the manuscript preparation. We sincerely apologize for these substantial errors. Following a thorough data re-analysis, corrections were applied to Supplemental Figure 10, including its legends and main-text description in the revised manuscript (please refer to the changes below).

Previous

Revised (color model was also revised)

206 simultaneously administrated into the medium. Five days later, the SA- β -gal staining was
207 performed in the VSMCs.↵
208 **(E-F)** SA- β -gal staining of VSMCs as indicated in the figure. **Ang II plus bleomycin (Ang II+Bleo)**
209 **were added into the culture medium to induce senescence for 5 days. Then, ROS scavenger**
210 **NAC (5 mM) or NAD⁺ precursor NMN (300 μ M) was then administrated into the medium for**
211 **additional 2 days.** The SA- β -gal staining was performed in the VSMCs.↵
212 Data expressed the mean \pm SEM. N = 6-8 per group. ** P <0.01. Comparisons of parameters
213 were performed with Unpaired t-test. NS, no significance. ↵

Supplemental Figure 10 legend

In main-text, the original sentence '*However, delayed administration of NAC or NMN, was unable to reverse the already-formed senescence driven by Ang II+Bleo or RSL3 (Supplemental Figure 10E-F)*' was wrong and has been changed to '*However, delayed administration of NAC or NMN, was unable to reverse the already-formed senescence driven by Ang II+Bleo (Supplemental Figure 10E-F)*'.

352 nicotinamide mononucleotide (NMN). Simultaneous treatment of NAC or NMN was able to
353 partially block the senescence induced by Ang II+Bleo and ferroptosis inducer RSL3
354 **(Supplemental Figure 10C-D)**. However, delayed administration of NAC or NMN, was unable
355 to reverse the already-formed senescence driven by **Ang II+Bleo (Supplemental Figure 10E-**
356 **F)**. These results indicate that **ferroptosis stress** drives senescence via secretome-dependent

Main-text (Line 355)

Q13. Supplementary Figure 11 - the newly supplied blot (and corresponding quantification) is undoubtedly more convincing and more consistent with the interpretation of the authors. However, how was this blot generated? From new samples? The adjacent blots (for NCOA4 and loading control H2B) have not changed/been updated??

Response: The newly supplied blot was rerun using the old samples stored at -80°C. Consequently, we did not change the adjacent blots. Thank you.

References

1. Victorelli, S. *et al.* Apoptotic stress causes mtDNA release during senescence and drives the SASP. *Nature* **622**, 627–636 (2023).
2. Fernandez-Duran, I. *et al.* Cytoplasmic innate immune sensing by the caspase-4 non-canonical inflammasome promotes cellular senescence. *Cell Death Differ* **29**, 1267–1282 (2022).

REVIEWERS' COMMENTS

Reviewer #4 (Remarks to the Author):

I'm happy with the author's rebuttal and changes for all points except for the following:

Q3

'we have replaced the term 'ferroptosis' with 'pro-ferroptotic signaling', 'ferroptosis stress', or 'ferroptosis activation' depending on the context '

-> The use of all of these terms to describe the same phenomena is confusing. Furthermore, 'ferroptosis stress' and 'ferroptosis activation' neither accurately capture the sentiment of the authors ('ferroptosis stress' implies stress as a consequence of ferroptotic cell death and 'ferroptosis activation' implies initiation of ferroptotic cell death), nor are these terms widely accepted or recognized in the context of ferroptosis, nor are the corresponding terms used for other forms of cell death e.g apoptosis and pyroptosis. The authors should consistently use either 'pro-ferroptotic signalling' (which best describes the signalling events that can lead to ferroptosis) or 'ferroptotic stress' (for consistency with the 'apoptotic stress' terminology used by Victorelli et al) throughout the manuscript.

'we incorporated a paragraph raising the issue of potential inherent contradiction between ferroptosis and senescence, along with our reflections on this matter (Lines 558-578). While our proficiency in English may not be entirely sufficient to convey our thoughts comprehensively, we sincerely hope that our revised version meets your expectations.'

-> This paragraph is important but currently cumbersome and would benefit from reworking. Suggestions include:

- remove 'In terms of the potential inherent contradictions between senescence ...ferroptosis' and start the

- paragraph with 'Our findings...'

- line 570, consider replacing 'complete' with 'irreversible' or 'permanent'

- lines 571 (starting with 'our work') to line 577 (ending in 'specific context') is repetition of lines 559-561 and can be removed.

Q7

(line 230) Please change 'ferroptosis' to 'pro-ferroptotic signalling' or 'ferroptotic stress' in accordance with response to Q3.

Point-to-point response to Reviewer #4

Q1. 'we have replaced the term 'ferroptosis' with 'pro-ferroptotic signaling', 'ferroptosis stress', or 'ferroptosis activation' depending on the context '. The use of all of these terms to describe the same phenomena is confusing. Furthermore, 'ferroptosis stress' and 'ferroptosis activation' neither accurately capture the sentiment of the authors ('ferroptosis stress' implies stress as a consequence of ferroptotic cell death and 'ferroptosis activation' implies initiation of ferroptotic cell death), nor are these terms widely accepted or recognized in the context of ferroptosis, nor are the corresponding terms used for other forms of cell death e.g apoptosis and pyroptosis. The authors should consistently use either 'pro-ferroptotic signalling' (which best describes the signalling events that can lead to ferroptosis) or 'ferroptotic stress' (for consistency with the 'apoptotic stress' terminology used by Victorelli et al) throughout the manuscript.

Response: We have replaced the terms 'ferroptosis stress' and 'ferroptosis activation' with the term 'pro-ferroptotic signalling' throughout the manuscript.

Q2 'we incorporated a paragraph raising the issue of potential inherent contradiction between ferroptosis and senescence, along with our reflections on this matter (Lines 558-578). While our proficiency in English may not be entirely sufficient to convey our thoughts comprehensively, we sincerely hope that our revised version meets your expectations.' This paragraph is important but currently cumbersome and would benefit from reworking.

Suggestions include:

- remove 'In terms of the potential inherent contradictions between senescence ...ferroptosis' and start the paragraph with 'Our findings...'

Response: According to this suggestion, we removed the sentence 'In terms of the potential inherent contradictions between senescence ...ferroptosis'.

- line 570, consider replacing 'complete' with 'irreversible' or 'permanent'

Response: According to this suggestion, we replaced the term 'complete' with 'irreversible'.

- lines 571 (starting with 'our work') to line 577 (ending in 'specific context') is repetition of lines 559-561 and can be removed.

Response: According to this suggestion, we removed these sentences.

- (line 230) Please change 'ferroptosis' to 'pro-ferroptotic signalling' or 'ferroptotic stress' in accordance with response to Q3.

Response: According to this suggestion, we have changed 'ferroptosis' to 'pro-ferroptotic signaling' throughout the manuscript.